# Identification and characterisation of serotonin signalling in the potato cyst nematode *Globodera pallida* reveals new targets for crop protection

Anna Crisford[1,◉], Fernando Calahorro[1,◉], Elizabeth Ludlow[1], Jessica M. C. Marvin[2], Jennifer K. Hibbard[2], Catherine J. Lilley[2], James Kearn[1,¤], Francesca Keefe[1], Peter Johnson[1], Rachael Harmer[1], Peter E. Urwin[2], Vincent O'Connor[1], Lindy Holden-Dye[1]*

**1** School of Biological Sciences, University of Southampton, Southampton, United Kingdom, **2** Centre for Plant Sciences, School of Biology, Faculty of Biological Sciences, University of Leeds, Leeds, United Kingdom

◉ These authors contributed equally to this work.
¤ Current address: Toxicology, Trauma & Medicine, dstl Porton Down, Salisbury, Wiltshire, United Kingdom
* lmhd@soton.ac.uk

**Data Availability Statement:** All relevant data are within the manuscript and its Supporting Information files.

## Abstract

Plant parasitic nematodes are microscopic pathogens that invade plant roots and cause extensive damage to crops. We have used a chemical biology approach to define mechanisms underpinning their parasitic behaviour: We discovered that reserpine, a plant alkaloid that inhibits the vesicular monoamine transporter (VMAT), potently impairs the ability of the potato cyst nematode *Globodera pallida* to enter the host plant root. We show this is due to an inhibition of serotonergic signalling that is essential for activation of the stylet which is used to access the host root. Prompted by this we identified core molecular components of *G. pallida* serotonin signalling encompassing the target of reserpine, VMAT; the synthetic enzyme for serotonin, tryptophan hydroxylase; the G protein coupled receptor SER-7 and the serotonin-gated chloride channel MOD-1. We cloned each of these molecular components and confirmed their functional identity by complementation of the corresponding *C. elegans* mutant thus mapping out serotonergic signalling in *G. pallida*. Complementary approaches testing the effect of chemical inhibitors of each of these signalling elements on discrete sub-behaviours required for parasitism and root invasion reinforce the critical role of serotonin. Thus, targeting the serotonin signalling pathway presents a promising new route to control plant parasitic nematodes.

## Author summary

Indian snakeroot is an herbal medicine prepared from the roots of the shrub *Rauwolfia serpentina* that has been used for centuries for its calming action. We have found that the major active constituent of snakeroot, reserpine, is a potentially powerful crop protectant

**Funding:** Anna Crisford and Elizabeth Ludlow were supported by Biotechnology and Biological Sciences (BBSRC) grant number BB/J006890/1. Fernando Calahorro was supported by an award from Bayer Grants4Targets. Jennifer Hibbard was supported by BBSRC grant number BB/J006017/1. Some C. elegans strains were provided by the CGC, which is funded by NIH Office of Research Infrastructure Programs (P40 OD010440). The funders had no role in study design, data collection and analysis, decision to publish, or preparation of the manuscript.

**Competing interests:** I have read the journal's policy and the authors of this manuscript have the following competing interests: Holden-Dye, O'Connor and Urwin are inventors for filed UK Patent Application no. 1710057.9 Reserpine Seed Coating; United States Patent Application No. 16/625,580 Reserpine Seed Coating; European Patent Application No. 187337401.2-1110/3641543 Reserpine Seed Coating.

as it protects the roots of plants from attack by microscopic plant parasitic nematodes. The pharmacology of reserpine is well known: it works by depleting a specific class of mood regulating chemical in the nervous system, the biogenic amines. We show that in the nematode, reserpine disables biogenic amine signalling and that loss of the amine serotonin halts the parasitic life cycle. We identified key components of the serotonin signalling pathway in the potato cyst nematode *Globodera pallida* and show that chemicals that target these sites also inhibit the ability of the nematode to invade its host plant. Therefore, the naturally occurring plant root alkaloid reserpine has provided insight into new targets for crop protection.

## Introduction

Plant parasitic nematodes (PPNs) are microscopic nematode worms that invade the roots of plants causing $125 billion of crop damage per annum [1]. Chemicals deployed to protect crops from PPNs are typically either nematostatics which paralyse the nematodes by acting as cholinesterase inhibitors or are metabolic poisons [2]. However, the off-target toxicity of these agents is proving unacceptable and they are being removed from use (e.g. EU regulation EC 1107/2009). This presents a growing economic burden that demands new approaches to crop protection. One of the ways to interfere with the infectivity of PPNs is to selectively and discretely disable behaviours that are intrinsic to their parasitic life cycle.

The subject of this investigation is the sedentary endoparasitic potato cyst nematode, of which there are two major, closely related species *Globodera pallida* and *Globodera rostochiensis*. Together these are important crop pests of worldwide economic significance [3,4]. The particular focus of this work is the white potato cyst nematode *G. pallida*, for which there is no single, dominant potato natural resistance gene available. It has a complex life cycle [5]: The parasitic cycle starts when second-stage juveniles (J2s) hatch from eggs and emerge from the cysts in close proximity to the host roots. These non-feeding juveniles are constrained by their energy stores and have limited time to locate and infect host plant roots. Once the roots are located J2s penetrate an epidermal cell, generally in the zone of elongation, and migrate intracellularly inside the roots to establish a feeding site. Feeding J2s progress through subsequent developmental stages with the vermiform adult males leaving the roots and adult females forming a cyst with hundreds of eggs enclosed within their tanned body wall. The nematode completes its cycle and the cysts are released into the soil where new nematodes will emerge in response to signals from suitable host roots in the next potato cropping season. As the eggs are protected and remain dormant in the cysts until favourable conditions for host plant invasion are established, this cycle can be repeated with a delay of 20 to 30 years. Thus, once a field is contaminated with *Globodera spp*. the threat to crops can be very enduring.

Disruption of the infectivity cycle at the earliest points possible is predicted to lead not only to a reduction in established nematodes and subsequent population build-up but, importantly, could prevent the root damage associated with the early destructive migration of J2s. This damage can reduce the rate of root growth and decrease rates of uptake for water and nutrients [6] likely contributing to reduced tolerance even amongst some resistant potato cultivars. The early target phase encompasses hatching, directed locomotion towards the host plant, detection of the host plant root and root invasion. An important component of these behaviours is the ability of the nematode to appropriately activate its stylet, a hollow lance-like structure that can protrude from its mouth, and is implicated in both hatching and penetration of the host plant root [7,8]. Once inside the host root, secretions from the nematode pharyngeal glands

are introduced via the stylet into a selected plant cell. These secretions are intimately involved in establishing and maintaining the syncytial feeding site from which the stylet itself then provides the sole route for nutrient acquisition. All of these behaviours are the output of the nematode's nervous system but little is currently understood of their neurobiology [8].

An informative approach is to test the effect of diagnostic pharmacological reagents on nematode behaviour and infectivity. It has previously been reported that compounds that interact with biogenic amine signalling, and in particular serotonin, elicit effects on stylet thrusting and reproductive behaviours [9–12]. Therefore, we focused on chemicals known to interfere with these signalling pathways. We report a remarkable effect of reserpine which we show potently elicits a long-lasting impairment of the ability of *G. pallida* J2s to invade the host plant root. Reserpine is a naturally occurring plant alkaloid from the shrub *Rauwolfia serpentina* known for centuries as an herbal medicine with a tranquillising action and introduced into the clinic as one of the earliest treatments for hypertension [13]. In the mammalian nervous system the molecular target of reserpine is well characterised and is the vesicular monamine transporter (VMAT) [14]: This transporter is required to store biogenic amine neurotransmitters in presynaptic vesicles in the nervous system and by blocking its activity reserpine depletes neurons of this core signalling capability. We show that in the nematode *G. pallida* the target of reserpine is also a VMAT and this is manifest as an inhibition of stylet thrusting thus preventing root invasion. This prompted us to use *Caenorhabditis elegans* to identify and functionally characterise further core molecular components of the serotonin signalling pathway in *G. pallida*; we report that targeting these with chemical inhibitors can also protect roots from infection. Given the universally important role of the stylet to the life cycle and feeding of PPNs we argue that targeting the core components of the serotonin signalling pathway presents an under-utilised and promising new route to control PPNs.

## Results

### The inhibitory effect of reserpine on *G. pallida* root invasion behaviours

Pre-incubation of *G. pallida* J2s with 100 μM reserpine prior to their inoculation onto potato hairy roots in tissue culture significantly decreased the number of nematodes present in the roots 13 days later (Fig 1A). In order to invade the host root the J2s must be motile and also must be capable of thrusting their stylet in a rhythmic manner to pierce the plant cell walls [5]. Therefore, we tested the effect of reserpine on *G. pallida* motility and stylet activity. For motility, we employed a dispersal assay in which J2s were placed in the centre of an agar arena, demarcated with six concentric rings, with the central origin labelled '0' and the outermost ring numbered '5'. The number of J2s present in each concentric ring after a given period of time was scored. Although J2s were still moving after 18 h pre-exposure to reserpine (50 μM) their dispersal was significantly reduced. After 1 h in the agar arena 27.8 ± 2.0% of J2s were at the origin compared to 63.4 ± 3.6% of the reserpine treatment group (n = 10 plates for each experimental group, from 2 independent repeats). This differential was sustained after the J2s had been on the agar arena for 2 h (Fig 1B). As reserpine is an inhibitor of the VMAT (14) (Fig 2A), which loads a range of biogenic amine neurotransmitters including serotonin and dopamine into their respective synaptic vesicular storage sites, the effect of reserpine on dispersal could be explained by depletion of any of these neurotransmitters.

In contrast, the biogenic amine serotonin is a known activator of stylet thrusting [9] which is a critical behaviour for root invasion for endoparasitic PPNs. Moreover, stylet thrusting is amenable to *in vitro* pharmacological investigation and therefore we investigated the mode of action of reserpine using this as a paradigm for root invasion behaviour. Fluoxetine, more commonly known as the antidepressant Prozac, was deployed to activate stylet thrusting of J2s

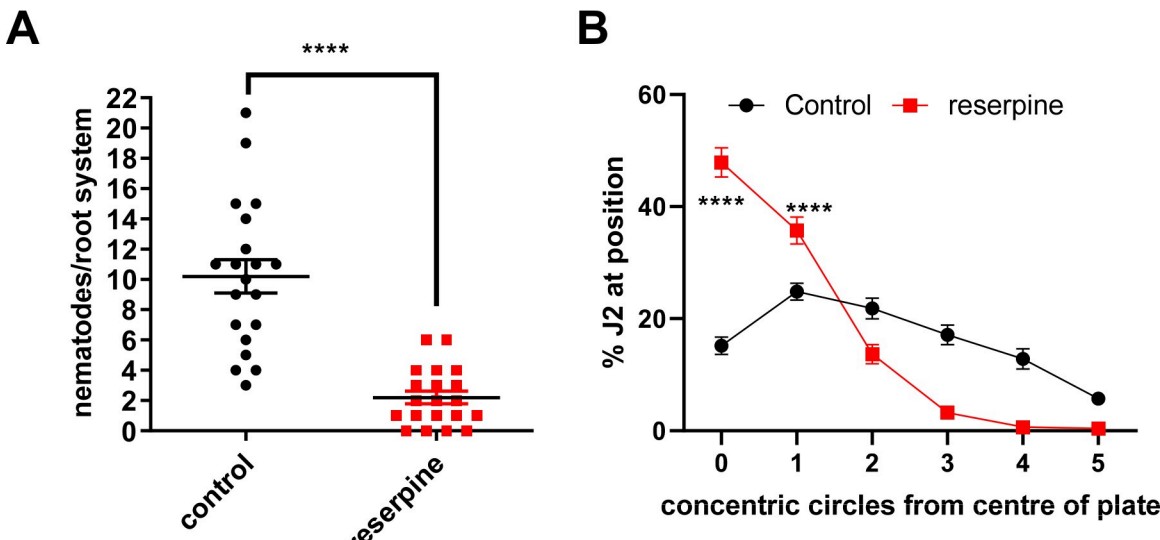

**Fig 1. Reserpine inhibits *G. pallida* host plant invasion behaviour. A.** J2s were collected at 24 h post hatching and pre-incubated with 24 h in water without (control), or with the addition of reserpine (100 μM). J2s were applied to individual potato hairy root cultures at 3 infection points with 25 J2s per infection point. 13 days later roots were stained with acid fuchsin and parasitic nematodes visualised. Data are mean ± s.e.m. for 20 plants from two independent experiments; **** p<0.0001 unpaired Student's t-test. **B.** The effect of reserpine on J2 motility was tested in a dispersal assay in which their ability to move away from the central point of an agar arena was determined. For each assay about 50 to 100 J2s were pre-soaked in water (control), or water with 50 μM reserpine for 18 h. They were then pipetted onto the centre of an agar plate in a minimum (~ 5 μL) volume of liquid. The plate was demarcated with concentric circles with the centre of the plate labelled '0' and the outermost circle as '5'. The percentage of J2s in each position was determined after 2 h. Data are mean ± s.e.m. of 10 plates for each experimental group from 2 independent experiments. There is a significant increase in J2 distribution at positions '0' and '1' with reserpine treatment compared to control ($F_{(5,54)}$ = 97.62, P<0.0001; two way ANOVA).

*in vitro* so that the inhibitory action of reserpine could be tested [10]. This compound selectively blocks the synaptic plasma membrane serotonin transporter. By preventing re-uptake of serotonin, fluoxetine increases serotonin concentration in the synaptic cleft which in turn activates the postsynaptic receptors (Fig 2A). Stylet thrusting can be visually scored by counting the frequency of projection/retraction cycles of the stylet (Fig 2B): Both serotonin and fluoxetine elicited a time-dependent and concentration-dependent stimulation of stylet thrusting. Responses reached a plateau after 30 min exposure (stylet thrusts for 2 mM fluoxetine after 30 min = 50 ± 11 $min^{-1}$, compared to 58 ± 9 $min^{-1}$ after 1 h, p>0.05: Stylet thrusts for 10 mM serotonin after 30 min = 54 ± 10 $min^{-1}$, compared to 58 ± 10 $min^{-1}$ after 1h, n = 17, p >0.05). The concentration–response curve for serotonin after 1 h reached a maximum plateau above 2 mM whilst fluoxetine had a bell-shaped concentration response curve (Fig 2C). This may indicate additional pharmacological sites of action for fluoxetine over and above the plasma membrane transporter for serotonin, most likely serotonin receptors. Serotonin receptor antagonism by fluoxetine at higher concentrations has been reported previously for *C. elegans* [15]. We tested the effect of reserpine against the maximally effective concentration of serotonin (10 mM) and fluoxetine (2 mM): Reserpine potently blocked the stylet response to fluoxetine but not the response to serotonin (Fig 2D) consistent with an interpretation in which the response to fluoxetine requires the presence of correctly stored vesicular serotonin which is depleted by the VMAT-blocking action of reserpine (see Fig 2A). We tested the reversibility of this reserpine inhibition by exposing *G. pallida* J2s to 50 μM reserpine with 2 mM fluoxetine for 10 h and then transferring them to reserpine free solution, with 2 mM fluoxetine, the 'recovery' period. Controls consisted of J2s that were exposed to 2 mM fluoxetine only. After 22 h in the recovery period stylet thrusting in the control group was 62.3 ± 6.5 $min^{-1}$ (n = 10)

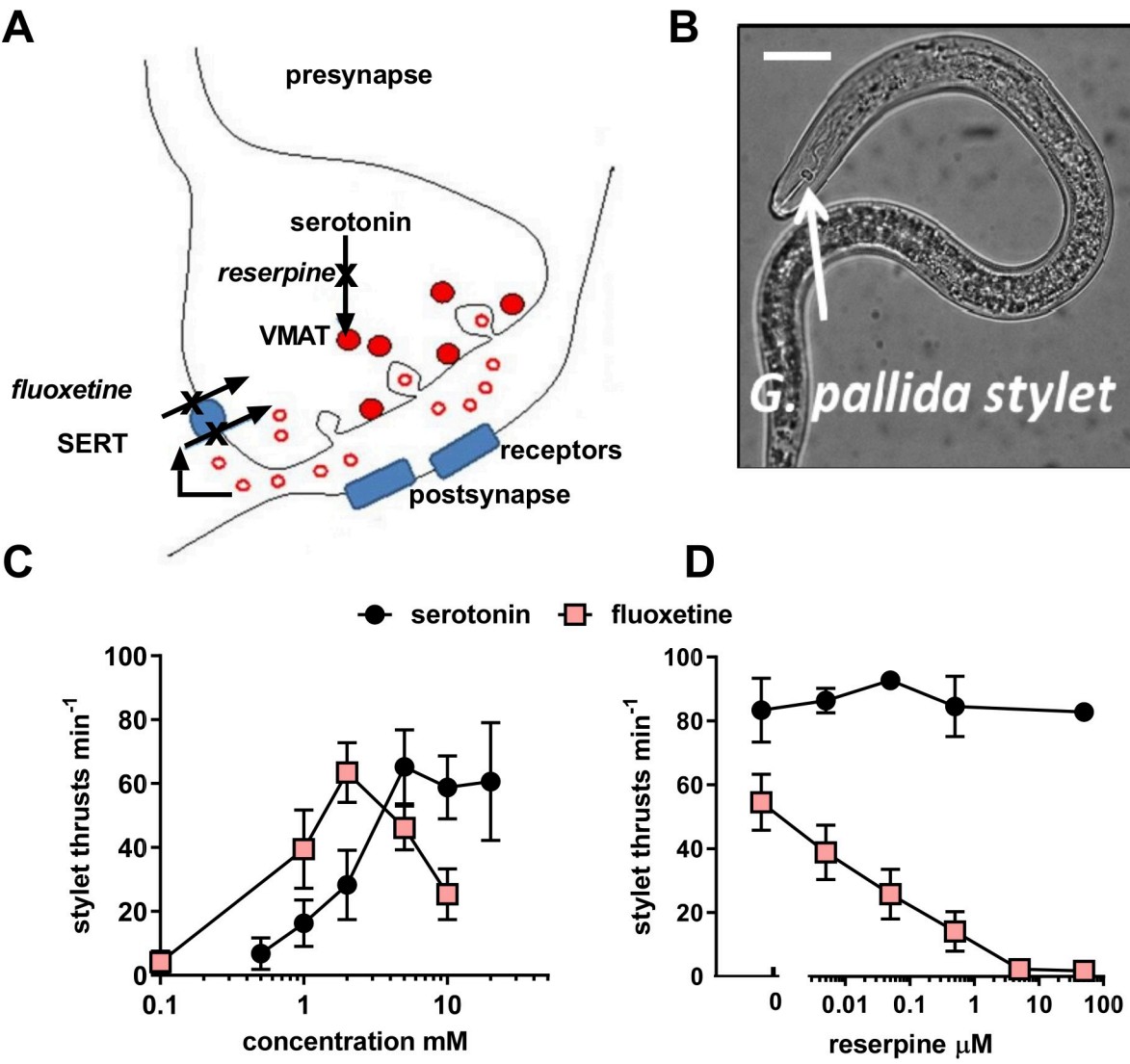

**Fig 2. Reserpine inhibits stylet thrusting triggered by endogenous, but not exogenous, serotonin. A.** The cartoon provides a rationale for the differential effect of reserpine on the response to serotonin and fluoxetine. It depicts a serotonergic synapse, with the pre-synapse as the site of serotonin synthesis and release and the post-synapse which harbours serotonin receptors. In mammalian systems reserpine acts presynaptically to deplete the storage of the neurotransmitter serotonin (shown as open red circles) by preventing its uptake into vesicles (shown as solid red circles) by inhibition of the vesicular monoamine transporter, VMAT, present on the vesicle membrane [14]. This prevents storage of serotonin and results in a lack of serotonin presynaptically and an inability of the presynaptic terminal to release serotonin. Fluoxetine (Prozac) on the other hand blocks the plasma membrane serotonin transporter, SERT. This prevents the reuptake of serotonin into the presynaptic terminal following its release. Thus, fluoxetine increases synaptic levels of serotonin (shown by open red circles) and in this way fluoxetine can act indirectly to stimulate transmission at the serotonergic synapse. This effect of fluoxetine is susceptible to block by reserpine as it requires endogenous serotonin to elicit its effect. However, the response to exogenous serotonin circumvents reserpine inhibition as it acts directly on the postsynaptic receptors. **B**. The *G. pallida* stylet is a lance-like structure that can be thrust out of the mouth of the nematode (in this image it is shown in the retracted position) in a rhythmic manner to initiate hatching and root invasion. Inside the root it is used for migration and to support feeding and interaction with the host. The activity of the stylet can be visually scored by counting the number of thrusts made in 1 min. Scale bar ~ 20 μm. **C.** J2s were incubated in either serotonin or fluoxetine at the concentrations indicated for 1 h and then the number of stylet thrusts made in 1 min was counted. Data are mean ± s.e.m.; n = 8 to 17 J2s for each time point. **D.** Reserpine blocked the stylet response to fluoxetine but not serotonin. J2s were pre-soaked in reserpine at the concentration indicated for 24 h and subsequently immersed in either 10 mM serotonin or 2 mM fluoxetine for 30 min and stylet thrusting scored for 1 min. Data are mean ± s.e.m.; n = 10 J2s for each data point.

whilst in the reserpine treated group it was $0.2 \pm 0.2$ min$^{-1}$ (n = 5; p<0.001). Thus, the inhibition of stylet thrusting by reserpine is sustained for at least 22 h following its removal.

Our observations do not provide a definition of the mechanism underpinning the inhibitory action of reserpine on J2 dispersal and this remains to be resolved. In contrast, it is clear that reserpine treatment leads to a sustained depletion of endogenous serotonin in neural circuits regulating stylet thrusting and this likely explains an inability of reserpine-treated nematodes to invade the host root. Furthermore, it indicates that serotonin signalling plays a key role in host plant invasion behaviour and therefore we embarked on a characterisation of core components of serotonergic neurotransmission with a view to testing these as potential targets for crop protection.

## *G. pallida* VMAT is a functional orthologue of *C. elegans* CAT-1

We identified the molecular target for reserpine in *G. pallida* using the sequence of the *C. elegans* vesicular monoamine transporter *cat-1* (WBGene00000295) which is well characterised [16] and for which there are two isoforms, *cat-1a* and *cat-1b*. We mined the *G. pallida* draft genome assembly [4] and identified a single *cat-1* homologue (GPLIN_000654600) which we designated *gpa-cat-1*. We cloned two variants of *G. pallida cat-1* from J2 cDNA, which had 2 and 3 amino acid differences, respectively, from the published sequence. These amino acid changes could not be assigned to any known functions of CAT-1 domains in *C. elegans* and are thus likely to be allelic variants. (S1 Fig).

In adult hermaphrodite *C. elegans cat-1* is expressed in all serotonergic and dopaminergic neurons [17]. *C. elegans cat-1(ok411)* functional null mutants exhibit a distinctive feeding phenotype; they cannot sustain fast pharyngeal pumping on food [16]. This reflects the major role that serotonin has in stimulating *C. elegans* pharyngeal pumping in the presence of food [18–20]. To test whether the sequence identified in *G. pallida* is a functional orthologue of *ce-cat-1* we expressed the *gpa-cat-1* cDNA in the null mutant *C. elegans* and tested for rescue of the *C. elegans* pharyngeal phenotype. The two cloned allelic variants of *gpa-cat-1* were both individually expressed in *C. elegans cat-1(ok411)* from a pan-neuronal promoter *Psnb-1* [21] and transgenic lines scored for the *cat-1 (ok411)* pharyngeal phenotype. This was compared to rescue achieved by expression of *C. elegans cat-1* from the same promoter i.e. *Psnb-1*. As *C. elegans cat-1a* is more similar to *G. pallida cat-1* this was used for the control rescue experiments. Transformed nematodes were identified by co-expression of *gfp* from the *myo-3* promoter which provides readily identifiable fluorescence in the body wall muscle of transgenic lines. The control nematodes expressing *myo-3*::*gfp* had the same pumping rate as untransformed *cat-1* mutants (*myo-3*::*gfp* $182 \pm 2$ and *cat-1* $192 \pm 2$ pumps min$^{-1}$, respectively; p>0.05) indicating expression of the transformation marker alone does not impact on pharyngeal pumping. Both *gpa-cat-1* and *C. elegans cat-1a* rescued the pharyngeal pumping phenotype of *cat-1 (ok411)* mutant nematodes (Fig 3A). For *C. elegans cat-1* this was a complete rescue as there was no significant difference in pumping rate between the wild type and rescue lines whilst for *G. pallida* the rescue was apparently partial as despite the restoration of pumping there remained a small but significant reduction in the rescue lines compared to N2. Thus *gpa-cat-1* encodes a vesicular monoamine transporter functional in *C. elegans*. The same rescue was observed using either *gpa-cat-1a* or *gpa-cat-1b* supporting the conclusion that both these sequences encode a functional VMAT in *G. pallida*.

The function of CAT-1 may also be assessed through the pharmacological response of the nematodes to fluoxetine. The impact of fluoxetine on pharyngeal pumping is not apparent in well fed *C. elegans* in the presence of food (bacteria) as the nematodes' pharyngeal activity is maximally activated by the presence of the bacteria. However, in the absence of bacteria

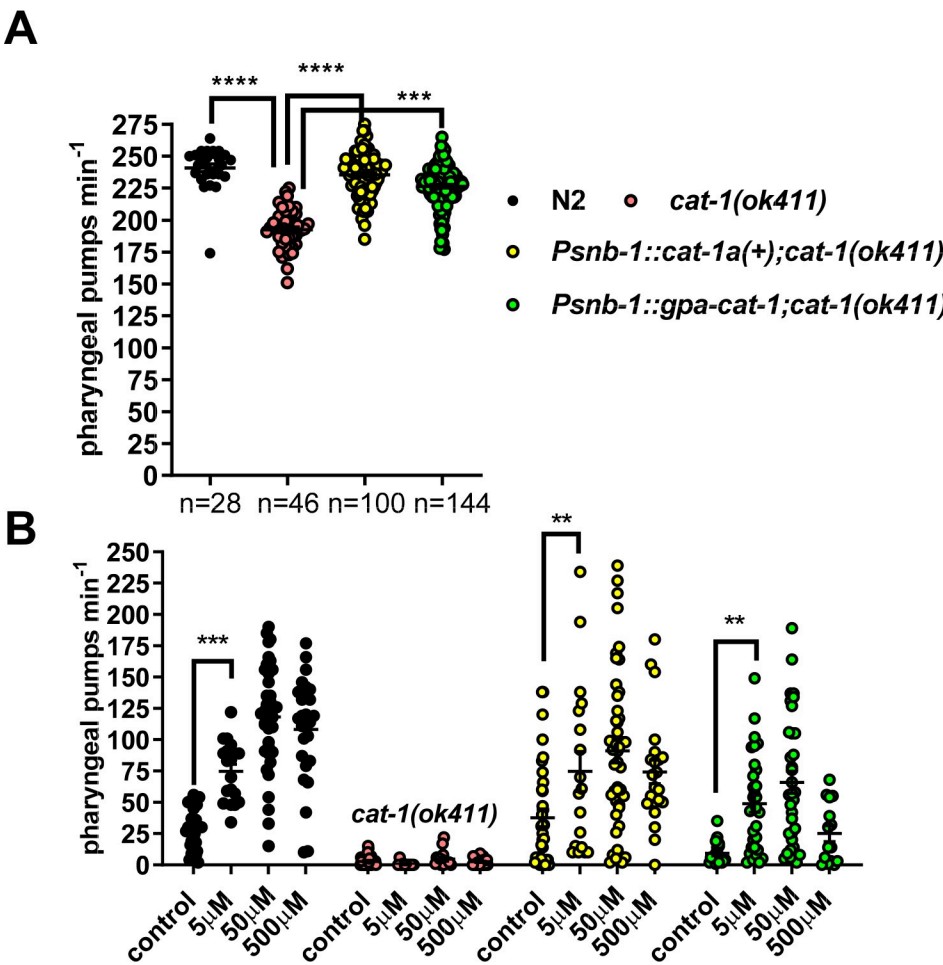

**Fig 3. Functional characterisation of a gene encoding *G. pallida* VMAT, *gpa-cat-1*. A.** Expression of *G. pallida cat-1* (*gpa-cat-1*) rescues the pharyngeal phenotype of *C. elegans cat-1(ok411)*. A comparison of the pumping rate on food for wild-type N2, *cat-1(ok411)* and transgenic lines of *cat-1(ok411)* expressing either wild type *C. elegans cat-1*, *cat-1(+)* or *gpa-cat-1* behind a pan-neuronal promoter (*Psnb-1*). Four stable lines for each transgenic strain were tested and the data are pooled for presentation. Data are mean ± s.e.m; 'n' is shown in brackets where 'n' is the number of *C. elegans*. *** p<0.001; **** p<0.0001; One way ANOVA with Bonferroni's multiple comparisons. There was no significant difference between N2 and *Psnb-1::cat-1a(+);cat-1(ok411)*, p = 0.7.84. There was a significant difference between N2 and *Psnb-1::gpa-cat-1;cat-1(ok411)* p = 0.002, indicating that the rescue for *G. pallida cat-1* is partial. **B.** Expression of *gpa-cat-1* in *C. elegans cat-1(ok411)* reinstated sensitivity to fluoxetine. One day old hermaphrodite *C. elegans* were placed on agar plates that either had no drug, 'control', or had been prepared with fluoxetine (5 to 500 μM). After 1 h the rate of pharyngeal pumping was scored in each nematode for 1 min. N2 wild type *C. elegans* pumped at a low rate which increased in a concentration-dependent manner in the presence of fluoxetine. Fluoxetine did not stimulate pumping in *cat-1(ok411)* but the response was restored in the transgenic *C. elegans* expressing either *cat-1(+)* or *gpa-cat-1* behind a pan-neuronal promoter (*psnb-1*). Two stable lines for each transgenic strain were tested and the data are pooled. Data are the mean ± s.e.m.; n ≥ 17 where 'n' is the number of *C. elegans*; **p<0.01; *** p<0.001; Two way ANOVA with Bonferroni's multiple comparisons. For the sake of clarity only one comparison is shown for each strain between control and 5 μM fluoxetine.

pharyngeal activity is much reduced: Under these conditions pharmacological stimulation of pharyngeal pumping with fluoxetine, which elevates synaptic serotonin, a key signal for activation of pumping [20,22], can be observed. The stimulatory effect of fluoxetine on *C. elegans* pharyngeal pumping in the absence of food was maximal after 1 h exposure and concentration-dependent (Fig 3B). Similar stimulation was not observed in the *C. elegans cat-1* mutant,

consistent with a model in which fluoxetine acts by elevating endogenous synaptic levels of serotonin by blocking reuptake via the serotonin transporter, SERT (*cf* Fig 2A)[23]. Higher concentrations of fluoxetine (1 mM and 2 mM) or longer exposure times at 500 μM, did not cause a stimulation of pharyngeal pumping which probably reflects that fact that at higher concentrations fluoxetine is an antagonist of serotonin receptors [15,19] thus any stimulatory action would be inhibited by concomitant receptor blockade. Importantly, with respect to establishing the function of *gpa-cat-1*, the low dose stimulatory effect of fluoxetine on pharyngeal pumping that was absent in the *C. elegans cat-1* mutant was restored by expression of either wild-type *C. elegans cat-1* or *gpa-cat-1* (Fig 3B). This confirms the identification of the vesicular monoamine transporter, VMAT, from *G. pallida*.

## Cloning and functional analysis of *G. pallida tph-1*

To further test the role of serotonin in the parasitism of PPNs we investigated the role of the synthetic enzyme for serotonin, tryptophan hydroxylase. The *C. elegans* gene encoding tryptophan hydroxylase *tph-1* (WBGene00006600) has two predicted isoforms, *tph-1a* and *tph-1b*. *C. elegans tph-1a* is assembled from 11 exons, while *tph-1b* has a shorter 5' terminus and is assembled from 9 exons. The *G. pallida* genome assembly contains a putative orthologue (GPLIN_000790300) that we designated *gpa-tph-1*. A single isoform was cloned from cDNA that more closely resembles the longer *ce-tph-1a* (S2 Fig).

A *C. elegans* functional null mutant for *tph-1(mg280)* has reduced pharyngeal pumping due to the lack of serotonin [17, 24]. To confirm the functional identity of *gpa-tph-1* as tryptophan hydroxylase we tested the ability of this gene to rescue the *C. elegans tph-1(mg280)* pharyngeal phenotype and compared this to the rescue obtained by expression of *C. elegans tph-1a(+)* both from a pan-neuronal promoter *Psnb-1*. Injection of the *tph-1* mutant with the transformation marker, *myo-3::gfp*, alone did not change pumping rate (*myo-3::gfp* 179 ± 4 pumps per minute; *tph-1(mg280)* 169 ± 3 pumps min$^{-1}$; p>0.05). *G. pallida tph-1* rescued the pumping phenotype of the *C. elegans tph-1(mg280)* mutant in a similar manner to expression of *C. elegans tph-1* (Fig 4A). For both the *C. elegans* and *G. pallida tph-1* rescue lines there was a small but significant difference between the rescue lines and wild type indicating the rescue is not complete. Taken together the observations support the conclusion that the *G. pallida* and *C. elegans tph-1* genes are functional orthologues.

We next wanted to investigate the functional importance of *G. pallida* TPH-1 in root invasion. Unfortunately, RNA interference through soaking in double-stranded (ds)RNA has non-specific toxic effects on *G. pallida* stylet thrusting confounding the use of this experimental approach [25]. Therefore, we investigated pharmacological blockers as a route to establish the functional involvement of serotonin signalling in *G. pallida* parasitism. We tested whether 4-chloro-DL-phenylalanine methyl ester hydrochloride (CPA), which is an inhibitor of the mammalian tryptophan hydroxylase [26], could be used as a chemical tool to test the function of TPH in *G. pallida*. As an inhibitor of TPH, we predicted that CPA would inhibit pharyngeal pumping in *C. elegans*. We found that pharyngeal pumping of wild type *C. elegans* was significantly inhibited by 2 h exposure CPA (10 mM) to a level that phenocopied the *tph-1* mutant and furthermore exposure of the mutant to CPA did not cause any further reduction in pharyngeal pumping (Fig 4B). The observation that the *tph-1* mutant occluded the ability of CPA to inhibit pharyngeal pumping supports the conclusion that CPA is a TPH-1 inhibitor in *C. elegans*. The ability of CPA to inhibit pharyngeal pumping was restored in *tph-1* mutants by expression of either *C. elegans* or *G. pallida tph-1* (Fig 4B). These data are consistent with CPA acting as a selective inhibitor of *C. elegans* and *G. pallida* TPH-1 and lent confidence to using CPA as a pharmacological tool to probe the role of TPH in *G. pallida* root invasion.

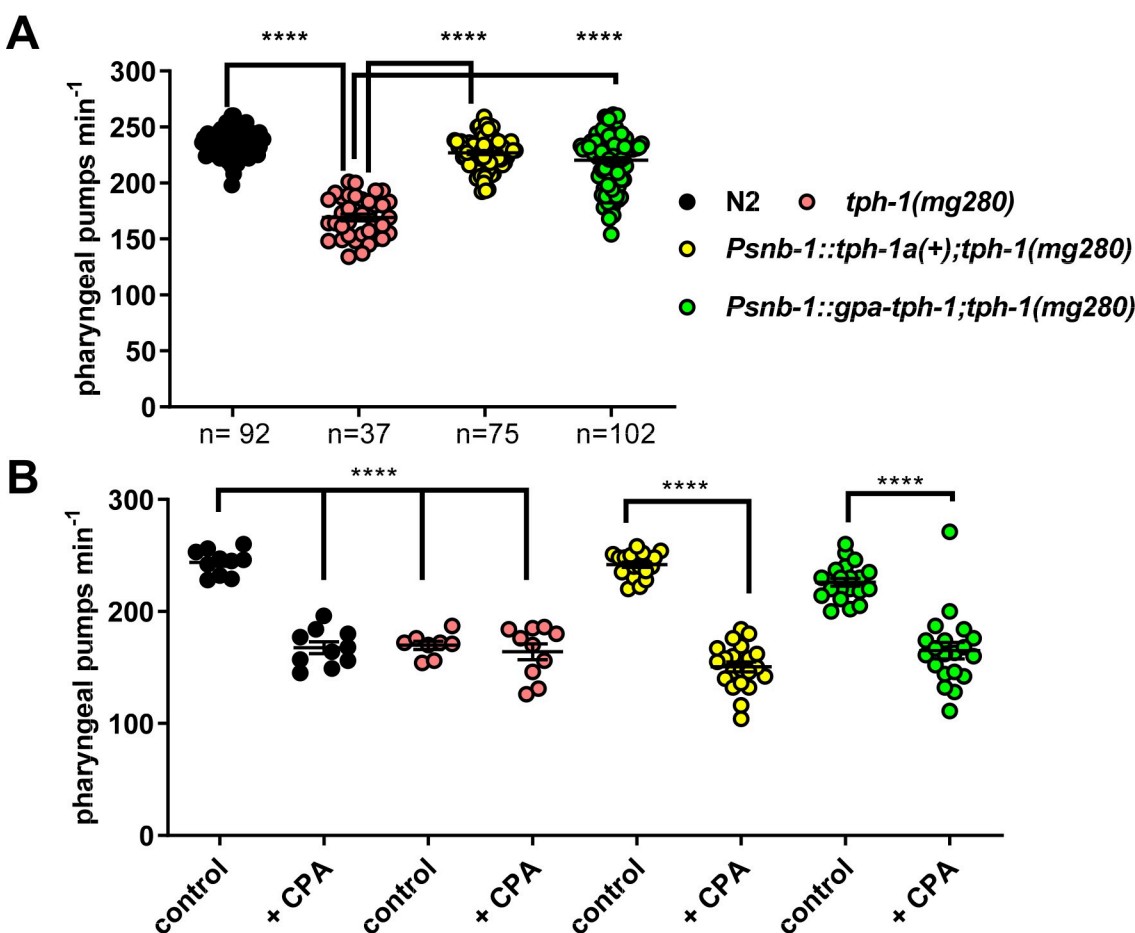

**Fig 4. Functional characterisation of a gene encoding *G. pallida* tryptophan hydroxylase, *gpa-tph-1*. A.** Expression of *gpa-tph-1* rescued the pharyngeal phenotype of *C. elegans tph-1(mg280)* similar to expression of a wild type copy of *C. elegans tph-1*, *tph-1(+)*. Both genes were expressed from the pan-neuronal promoter *Psnb-1*. Pharyngeal pumping rate of one day old adult hermaphrodites on food was scored for 1 min for each nematode. Four stable lines for each transgenic strain were tested and the data are pooled. Data are mean ± s.e.m; 'n' is indicated on the graph where 'n' is the number of *C. elegans*; **** p<0.0001; one way ANOVA with Bonferroni's multiple comparisons. There was a significant difference between N2 and *Psnb-1::C. elegans tph-1* (p = 0.008) and between *Psnb-1::G. pallida tph-1* (p = 0.001) indicating the rescue is not complete. **B.** Expression of either *C. elegans tph-1(+)* or *gpa-tph-1* in the *C. elegans* mutant *tph-1(mg280)* conferred sensitivity to the tryptophan hydroxylase inhibitor, CPA. One day old adult hermaphrodites were incubated for 2 h on agar plates with food (control), or with food and 10 mM CPA. Two stable lines were tested for each transgenic strain and the data are pooled. Data are the mean ± SEM of n ≥ 10 where 'n' is the number of *C. elegans*; ***p<0.001; one way ANOVA with Bonferroni's multiple comparisons.

## Targeting TPH-1 impairs *G. pallida* root invasion behaviour

We used CPA to test the role of the enzyme encoded by *gpa-tph-1* in root invasion behaviour of *G. pallida*. *G. pallida* J2s were treated with 10 mM CPA for 24 h after which the treated and control nematodes were soaked in serotonin (10 mM) or fluoxetine (2 mM) for 30 min and stylet thrusting visually scored. J2s incubated in CPA had reduced stylet thrusting in response to fluoxetine but maintained their response to serotonin suggesting that CPA inhibited TPH-1 enzyme activity and thus serotonin synthesis (Fig 5A) and consequently the response to fluoxetine which is dependent on endogenous serotonin. Treatment of J2s with CPA (100 µM) for 24 h prior to infection significantly reduced the number of nematodes present in potato roots 13 days after inoculation (Fig 5B) adding further evidence to support a role for serotonin signalling in root invasion.

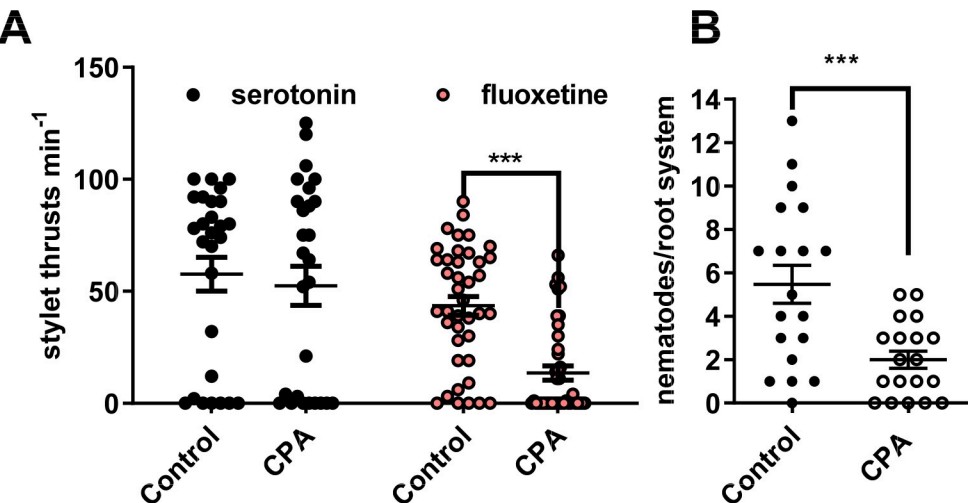

**Fig 5. The role of TPH-1 in host plant invasion behaviour. A.** *G. pallida* J2s were soaked in either water (control) or water with the TPH inhibitor CPA (10 mM) for 24 h. This was followed by the addition of either serotonin (10 mM) or fluoxetine (2 mM) for 30 min and stylet thrusting was scored for 1 min. Data are mean ± s.e.m. n = 27 for serotonin and n = 40 for fluoxetine. *** p<0.001; Two way ANOVA with Bonferroni's multiple comparisons. **B.** CPA impaired the ability of J2s in root invasion. J2s were collected at 24 h post hatching and pre-incubated for 24 h in water without (control), or with the addition of CPA (100 µM). J2s were applied to individual potato hairy root cultures at 3 infection points with 25 J2s per infection point. 13 days later roots were stained with acid fuchsin and nematodes visualised. Data are mean ± s.e.m. for 19 plants; *** p<0.001 unpaired Student's t-test.

## Functional and pharmacological characterisation of *G. pallida* serotonin receptors

The characterisation of *G. pallida* VMAT and TPH described above strongly supports a role for serotonergic transmission in host root invasion and provides a justification for pursuing the identity of *G. pallida* serotonin receptors that are involved in this behaviour. For this we used the well characterized serotonergic signalling pathway in *C. elegans* as our reference point [27–31]. Similar to their closest homologues in humans, *C. elegans* SER-1 and SER-4 have a low affinity for serotonin whilst SER-7 has a high affinity. Interestingly, MOD-1, which has a high affinity for serotonin, is a serotonin-gated chloride channel which is selectively found in the invertebrate phyla [32]. *G. pallida* putative orthologues of the *C. elegans* genes encoding serotonin receptors were identified by reciprocal best BLAST hit analysis of the *G. pallida* predicted gene complement. This identified *G. pallida* genes encoding candidate orthologues for the *C. elegans* receptors SER-4, SER-7 and MOD-1 although clear orthologues of SER-1 and SER-5 were not found (Table 1). In this study we focused on characterisation of SER-7, selected because of its key role in regulating feeding behaviour in *C. elegans* [19] and MOD-1, selected because of its important role in regulating locomotion underpinning exploration [33] in the anticipation that these receptors may sub-serve similar behaviours in plant parasitic nematodes.

SER-7 is a key determinant of *C. elegans* pharyngeal pumping [19,22,29]. This is demonstrated by the observation that *C. elegans* null mutants for *ser-7* display irregular pumping on food and have a reduced response to serotonin-stimulated pumping in the absence of food [19]. We cloned the closest homologue of *C. elegans* ser-7 from *G. pallida* (S3 Fig). To determine if this *G. pallida* sequence encodes a serotonin receptor, we tested its ability to rescue the lack of serotonin- stimulated pharyngeal pumping in the *C. elegans ser-7* mutant. We found that expression of either the putative *G. pallida* ser-7 or *C. elegans* ser-7 from the pan-neuronal promoter *snb-1* restored sensitivity of the pharyngeal system to serotonin (Fig 6A). In these *in*

**Table 1. Identification of *C. elegans* serotonin receptor homologues in *Globodera* spp.** *Globodera pallida* and *G. rostochiensis* homologues of the five *C. elegans* genes encoding serotonin receptors were identified by reciprocal best BLAST hit analysis of the *G. pallida* [4] and *G. rostochiensis* [3] predicted gene complements.* All are G-protein coupled receptors with the exception of MOD-1 which is a serotonin-gated chloride channel. The likely *gpa-ser-1* orthologue is incomplete in the genome assembly and encompassed by two adjacent gene models; this was confirmed by analysis of a transcriptome assembly produced using existing J2 RNAseq data [4]. # The complete sequence for *gpa-ser-5* was not present in the genome assembly, but was identified from the transcriptome assembly.

| *C. elegans* receptor | *C. elegans* isoforms | Potential orthologue in *G. pallida* | *G. pallida* amino acid identity (%) to *C. elegans* | Potential orthologues in *G. rostochiensis* | *G. rostochiensis* amino acid identity (%) to *C. elegans* |
|---|---|---|---|---|---|
| SER-1 | *ser-1a, ser-1b* | GPLIN_000650900/ GPLIN_000651000 | 26.7 | GROS_g04519 | 28.4 |
| SER-4 | | GPLIN_000001100 | 41.2 | GROS_g07000 | 47.0 |
| SER-5 | | YES # | 31.6 | GROS_g05497 | 29.6 |
| SER-7 | *ser-7a, ser-7b, ser-7c* | GPLIN_000340600 | 33.2 | GROS_g07752 | 35.3 |
| MOD-1 | *mod-1a, mod-1b, mod-1c* | GPLIN_001254300 | 46.1 | GROS_g02866 | 45.8 |

*vivo* whole organism experiments a high concentration of serotonin is required to activate the pharynx as the nematode's cuticle presents a permeability barrier to drug access [34] confounding discrete pharmacological analysis of the SER-7 receptor. Moreover, although SER-7 is an important contributor to the pharyngeal serotonin response in the intact nematode it is not the sole receptor involved, as evidenced by the residual response to serotonin in the *ser-7* mutant (Fig 6A). To circumvent these confounds we analysed the pharyngeal response to serotonin in a cut-head assay in which the intact pharynx is carefully cut from the rest of the nematode (Fig 6B). This enables drugs to access the tissue without having to cross the nematode cuticle and permits more precise determination of the concentration-dependence of the compounds' effects on pharyngeal activity [20,35]. Furthermore, the response of the pharynx in the cut-head preparation to serotonin is entirely dependent on SER-7 (Fig 6C). The isolated pharynx has previously been shown to be 2 to 3 orders of magnitude more sensitive to the stimulatory action of serotonin compared to the intact nematode [20]. Consistent with this, wild type nematodes showed a concentration-dependent increase in pharyngeal pumping in response to serotonin (EC$_{50}$ 223 nM, 95% confidence from 164 nM to 304 nM) whilst s*er-7(tm1325)* mutants did not respond even to 100 μM serotonin (Fig 6C). This is in agreement with earlier studies showing SER-7 is the key receptor intrinsic to the pharyngeal system that mediates serotonin activation of feeding [19]. Pharyngeal pumping in transgenic nematodes expressing either *psnb-1::ser-7a(+)* or *psnb-1::gpa-ser-7* was activated by serotonin with an EC$_{50}$ of 782 nM (95% confidence from 457nM to 1.34 μM) and EC$_{50}$ 3.9 μM (95% confidence from 2.16 μM to 7.08 μM), respectively (Fig 6C). We used this cut head pharmacological assay to test a putative antagonist of the SER-7 receptor, methiothepin [29] and showed that this compound blocked the stimulatory effect of serotonin in transgenics expressing either *C. elegans ser-7(+)* or *G. pallida ser-7* (Fig 6D) with around 50% inhibition at 100 nM. Together, this supports the conclusion that *gpa-ser-7* encodes a *G. pallida* serotonin receptor which has the closest similarity to *C. elegans* SER-7.

We next cloned a putative orthologue of the *C. elegans* serotonin-gated chloride channel *mod-1* from *G. pallida* (S4 Fig). This receptor belongs to a family of biogenic amine-gated chloride channels which are unique to the invertebrate phyla and thus of particular interest with respect to the development of novel chemical control agents that have potential for more limited off target toxicity.

We found that a null mutant for *C. elegans mod-1*, *(ok103)*, has a slightly reduced rate of pharyngeal pumping compared to controls. The pumping rate on food for *mod-1(ok103)* was 214 ± 3 min$^{-1}$ (n = 10) and in the control injected line (*myo-3::gfp;mod-1(ok103)*) 189 ± 7 min$^{-1}$

 

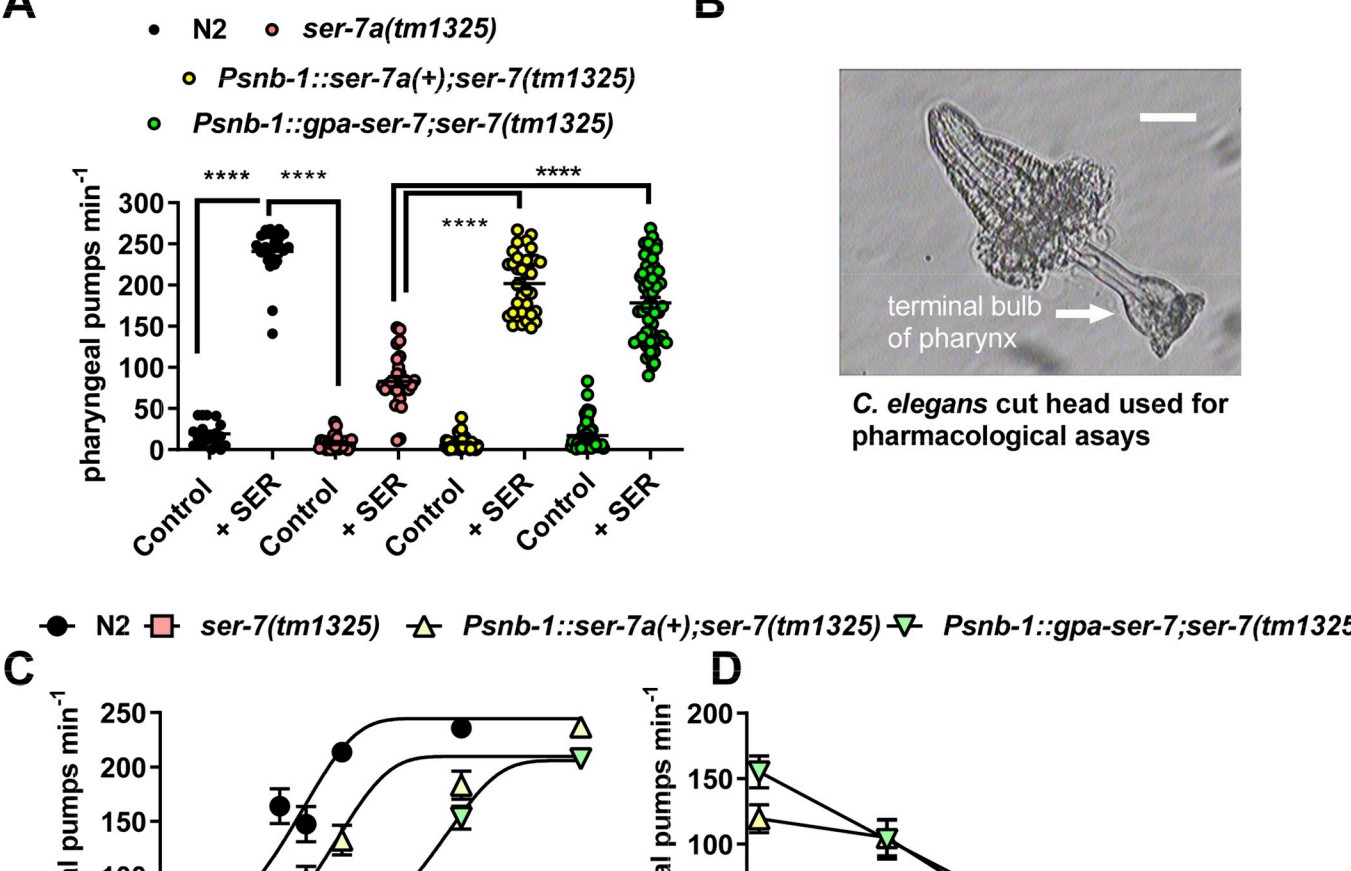

**Fig 6. Characterisation of *G. pallida* SER-7 A.** Measurements of pharyngeal pumping in the intact nematode: Expression of *gpa-ser-7* rescued the pharyngeal phenotype of *C. elegans ser-7(tm1325)* similar to expression of a wild type copy of *C. elegans ser-7a*, *ser-7a(+)*. Both genes were expressed from the pan-neuronal promoter *Psnb-1*. Pharyngeal pumping rate of one day old adult hermaphrodites after 20 min off food was scored for 1 min for each nematode either in the absence (control) or presence (SER) of 10 mM serotonin. Three stable lines for each transgenic strain were tested and the data are pooled. Data are mean ± s.e.m; n ≥ 20; **** p<0.0001; one way ANOVA with Bonferroni's multiple comparisons. Note that whilst the pumping rate of *ser-7(tm1325)* was increased by 10 mM serotonin this response was significantly lower than that for wild type and both transgenic strains.The stimulation of pumping rate by serotonin in the rescue lines *Psnb-1*::*C. elegans ser-7* and *Psnb-1*::*G. pallida ser-7* was significantly different from both N2 (p = 0.0001) and from *ser-7* (p = 0.001). **B.** An image of the cut head preparation that was used for the experiments shown in C and D. Cutting the head exposes the pharynx to the external solution allowing ready access of applied drugs to the receptors regulating the pharyngeal network. Pharyngeal pumps were recorded visually by counting the contraction-relaxation cycles of the terminal bulb. Scale bar ~25 μm. **C.** Concentration-response of the pumping rate of the cut head pharyngeal preparations to serotonin for wild type (N2), *ser-7 (tm1325)* and transgenic lines of *ser-7(tm1325)* expressing either *C. elegans* or *G. pallida ser-7* behind a pan-neuronal promoter (*psnb-1*). Note that in the dissected pharyngeal preparation *ser-7(tm1325)* mutants do not respond to even 100 μM serotonin. Data are the mean ± s.e.m.; n ≥ 20. **D.** Methiothepin (MET) blocked the response to serotonin in both transgenic strains. Cut heads were pre-incubated with methiothepin at the concentrations indicated for 5 min after which 100 μM serotonin was added and after a further 10 min the pumping rate was scored for 1 min. Two stable lines for each transgenic strain were tested and the data are pooled. Data are the mean ± s.e.m; n ≥ 5.

(n = 12) compared to 252 ± 4 (n = 7) in wild-type (p<0.0001, one way ANOVA with Bonferroni's multiple comparisons). This effect on pharyngeal pumping was rescued in lines expressing *C. elegans mod-1(+)* from the pan-neuronal promoter *Psnb-1* (230 ± 7 min⁻¹; n = 9; p<0.0001). This is consistent with a role for MOD-1 in regulating *C. elegans* feeding behaviour. However, a

more distinctive phenotype of *C. elegans mod-1 (ok103)* is that it does not exhibit the acute transient paralysis induced in wild type by a high concentration (33 mM) of serotonin [31]. We found that the putative *G. pallida* sequence encoding the orthologue of the *C. elegans* MOD-1 receptor could completely rescue this serotonin-induced phenotype in *mod-1(ok103)* as robustly as the expression of wild type *C. elegans mod-1* (Fig 7A). For these experiments we cloned the putative native *C. elegans mod-1* promoter to drive expression of both the *G. pallida* and *C. elegans* sequences. Like SER-7, *C. elegans* MOD-1 is also blocked by methiothepin therefore to further characterise the *G. pallida* receptor we compared the ability of methiothepin to block *C. elegans* and *G. pallida* MOD-1. We showed that the *mod-1* dependent serotonin-induced paralysis in both wild type *C. elegans* and in *C. elegans* expressing *G. pallida mod-1* was blocked by methiothepin (Fig 7B). This supports the conclusion that the *G. pallida* sequence encodes a functional orthologue of the *C. elegans* serotonin-gated chloride channel MOD-1.

In view of the evidence that a low micromolar concentration of methiothepin can block both *G. pallida* SER-7 and MOD-1 we speculated that this compound might provide insight into whether either, or both, of these receptors are involved in host plant invasion behaviour. We found that methiothepin is a remarkably potent antagonist of serotonin-induced stylet thrusting in *G. pallida* J2s (Fig 8A) and blocked root invasion (Fig 8B). Interestingly, stylet thrusting is also implicated in egg hatching [5] and we found that methiothepin potently inhibited the emergence of J2s from eggs (Fig 8C) although the J2s inside the eggs appeared viable (Fig 8D). In contrast, exposure of *G. pallida* cysts to 50 μM reserpine did not inhibit hatching (n = 20 cysts). This may reflect an inability of reserpine to penetrate the cyst casing to exert an inhibitory effect.

## Discussion

The most economically important PPNs are the cyst (e.g. *Heterodera* and *Globodera* spp.) and root-knot nematodes (*Meloidogyne* spp.). These pests present an increasing threat to global food security which is exacerbated by the imposition of restraints on the use of nematicides due to their unacceptable levels of hazard and ecotoxicity. Strategies to improve this situation are highly dependent on achieving a better understanding of the biology underpinning the interaction of these nematodes with their host plants [36,37].

There are discrete differences in the life cycle and behaviour of different species of PPNs but for all of them the ability to regulate their motility and the thrusting behaviour of their stylet in response to discrete environmental cues is core to their parasitic potential. In this study we have improved understanding of the neurobiological basis of these behaviours and shown a critical role for core components of a serotonergic signalling pathway. Our route into this was *via* a chemical biology approach in which we investigated the ability of diagnostic neuroactive drugs to impact on root invasion behaviour. We included the biogenic amine depleting compound reserpine in view of previous studies that have implicated biogenic amine transmitters, in particular serotonin, in PPN motility and stylet thrusting [9]. The molecular mode of action lies in its potent inhibition of the vesicular monoamine transporter, VMAT [14] which elicits a global depletion of biogenic amines in the nervous system. Reserpine is one of the bioactive constituents in root extracts of *Rauwolfia serpentina*, commonly called Indian snakeroot, which have been used in herbal medicine for centuries for their sedative properties. Last century reserpine was one of the first drugs to be registered for the treatment of hypertension [38] and was also used as an antipsychotic [39]. We found that pre-treatment of *G. pallida* J2s with reserpine impaired their ability to invade the host root and this was accompanied by a reduction in motility which could contribute to this potential crop protecting effect. Cross-referencing the effective concentration of reserpine to the effective concentrations of the major

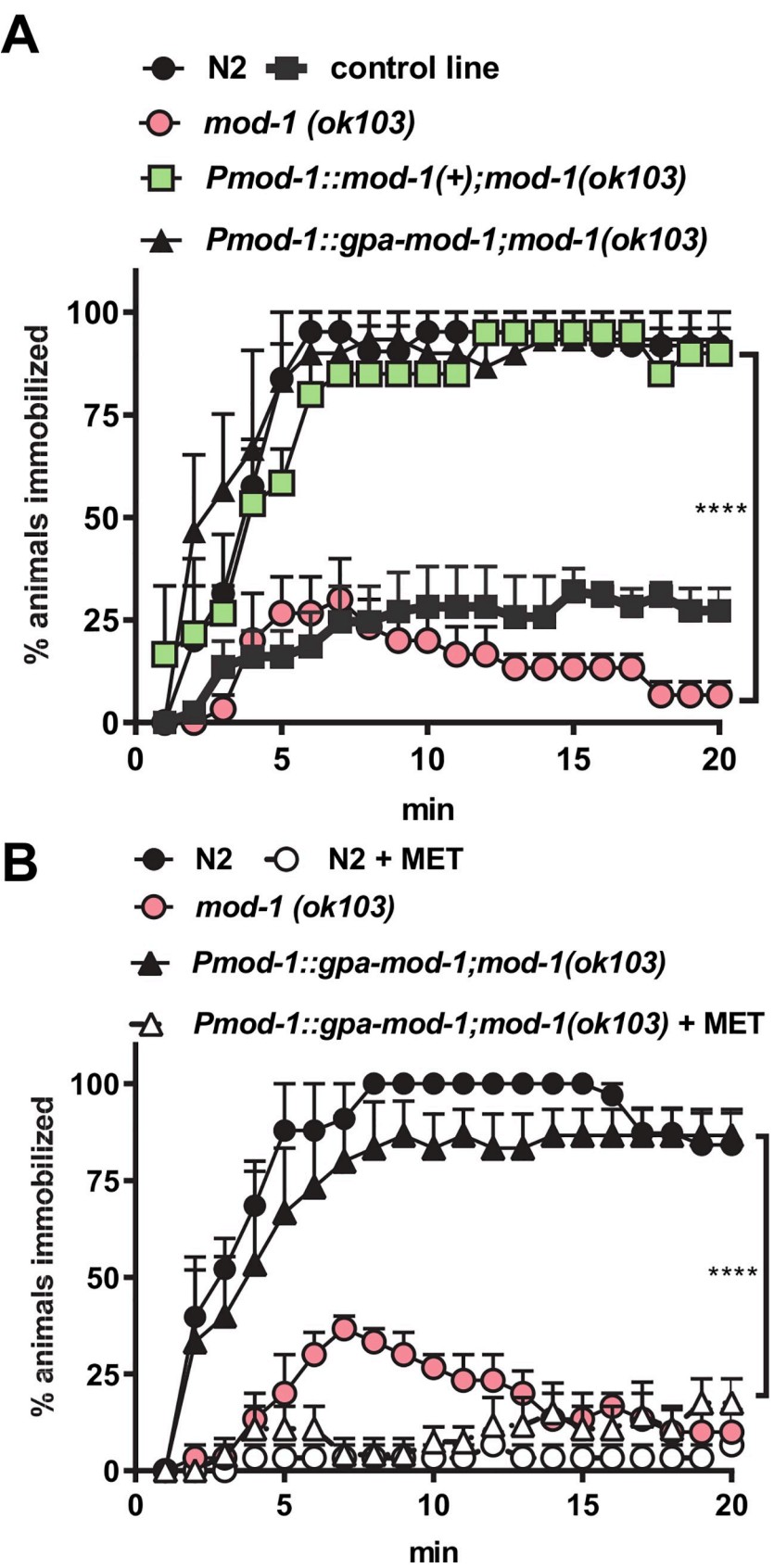

**Fig 7. Characterisation of *G. pallida* MOD-1. A.** Expression of *gpa-mod-1* rescued the motility phenotype of *C. elegans mod-1(ok103)* similar to expression of a wild type copy of *C. elegans mod-1, mod-1(+)*. The control expressed the transformation marker only. Both genes were expressed from the putative native *mod-1* promoter, *Pmod-1*. Approximately 10 one day old adult hermaphrodite *C. elegans* were dispensed into each well of a 96-well microtitre plate. At time 0, 33 mM serotonin dissolved in M9 buffer was added to each well and the percentage of immobilised nematodes in each well was scored every min for up to 20 min. Three stable lines for each transgenic strain were tested and the data are pooled. Data are mean ± s.e.m; n = 3 independent experiments; **** p<0.01; two way ANOVA with Bonferroni's multiple comparisons.There was no significant difference between N2 and either *Pmod-1::C. elegans mod-1* or *Pmod-1::G. pallida mod-1*. **B.** The serotonin-induced paralysis was blocked by methiothepin (MET) in both wild type (N2) and in the transgenic strain expressing *gpa-mod-1* in *mod-1(ok103)*. The experiment was performed as in A with the addition of a preincubation in 10 μM MET on an agar plate seeded with OP50 for 2 h and the inclusion of 10 μM MET with serotonin in the microtitre plate wells. Data are mean ± s.e.m; n = 3 independent experiments; ** p<0.01; two way ANOVA with Bonferroni's multiple comparisons.

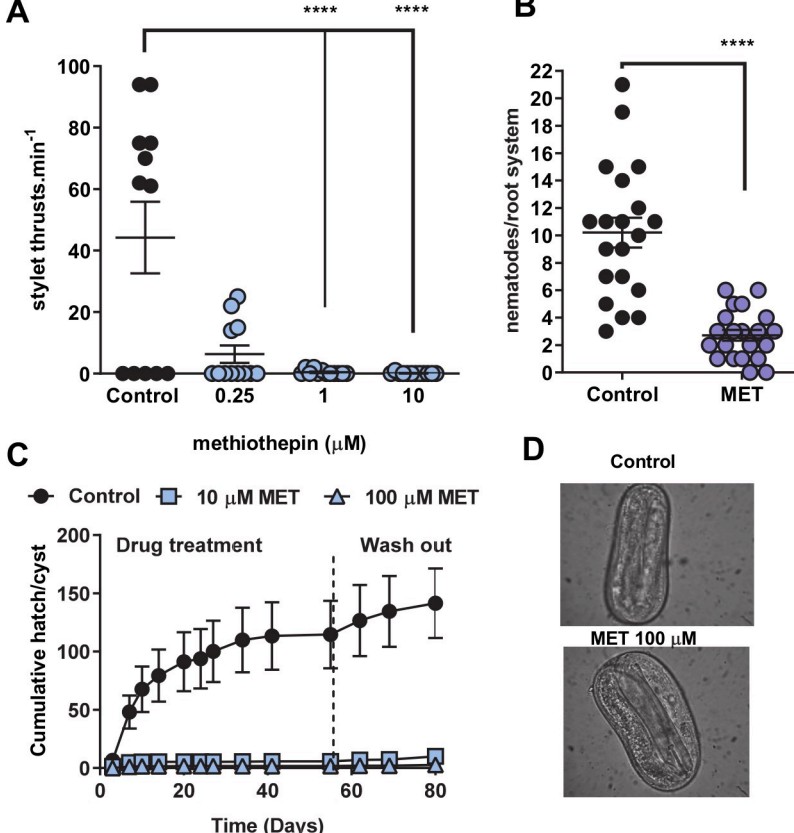

**Fig 8. Methiothepin is a potent inhibitor of *G. pallida* behaviours, hatching and root invasion, that are dependent on serotonin regulated stylet thrusting. A.** *G. pallida* J2s were soaked in either a control solution of 0.5% ethanol or methiothepin at different concentrations for 24 h. They were transferred to 2 mM fluoxetine and after 30 min had elapsed the rate of stylet thrusting was counted for 1 min. Data are mean ± s.e.m.; n = 12; p<0.0001; one-way ANOVA with Bonferroni's multiple comparisons. **B**. Methiothepin (MET) impaired the ability of J2s to invade the host root. J2s were collected at 24 h post hatch and pre-incubated for 24 h in water without (control), or with the addition of MET (100 μM). J2s were applied to individual potato hairy root cultures at 3 infection points with 25 J2s per infection point. 13 days later roots were fuschin stained and visualised for J2. Data are mean ± s.e.m. for 20 plants for control and 21 plants for MET; **** p<0.001 unpaired Student's t-test. **C.** MET blocked hatching of J2s from cysts. *G. pallida* cysts were soaked in a 24-well plate with 1 cyst per well in the presence of potato root diffusate modified with either vehicle (PRD), 10 μM methiothepin (M) or 100 μM methiothepin and hatching of J2 juveniles was scored. At 24 days, the cysts were placed into PRD alone to assess recovery of hatching. Methiothepin inhibited hatching and hatching did not recover on removal from drug (P<0.0001). **D.** Representative images of unhatched eggs from cysts treated with a control solution and 100 μM methiothepin. Note that methiothepin treatment does not appear to have negatively impacted the integrity of the unhatched J2 (egg dimensions 120 μm by 60 μm).

nematicidal compounds e.g. aldicarb and abamectin, shows that reserpine is at least as, if not more, efficacious [40].

To get a better insight into the neuropharmacological basis of the inhibitory action of reserpine on root invasion we developed a bioassay for monitoring stylet thrusting [41,42]. To assess the inhibitory effect of reserpine we first activated stylet thrusting by addition of either exogenous serotonin or fluoxetine. Serotonin induces rhythmic activity of the stylet and likely acts in a pathway that couples sensory perception of environmental cues to downstream signalling pathways to activate, or possibly disinhibit, stylet thrusting. This has parallels with *C. elegans* which also deploys serotonin signalling in response to environmental food cues to activate rhythmic pumping of its feeding organ, the pharynx [18,22]. We found that fluoxetine also induces stylet thrusting and this activation depends on the availability of endogenous serotonin that accumulates in the context of inhibition of uptake. Studies in *C. elegans* indicate this may be explained by an inhibition of serotonin reuptake [43] from the synapse and therefore increased levels of serotonin available to activate synaptic receptors. This approach therefore allowed us to test the hypothesis that reserpine exerts its effect by depleting endogenous serotonin. Interestingly, the concentration-response curve to fluoxetine was bell-shaped with lower stimulation observed at the highest concentrations tested. This may be explained by a direct receptor blocking effect of fluoxetine at high concentrations, a phenomenon also reported for *C. elegans* [15,43]. Notably, we found that reserpine exerted a selective and potent inhibition of stylet thrusting elicited by fluoxetine but not by serotonin indicating reserpine exerts its action by depleting endogenous serotonin. The estimated $EC_{50}$ for this effect of reserpine was 10 nM showing it has an extremely potent action, comparable to its effect on the mammalian VMAT1 and VMAT2 transporters [14]. Moreover, as might be predicted from a compound with this mode of action, the reserpine inhibition of fluoxetine-stimulated stylet thrusting was sustained and there was no recovery after 22 h removal from reserpine.

The serotonergic signalling pathway in PPNs is therefore a good candidate for new targets for crop protection. However, the molecular components of this have not previously been characterised despite the availability of PPN sequenced genomes [3,44]. In contrast, in *C. elegans* each element of the serotonergic pathway, from the synthetic enzymes through to the receptors has been identified and functionally assessed through the phenotypic delineation of their respective mutant strains. We made use of this to identify candidate orthologues of the respective *C. elegans* genes in the *G. pallida* genome. This revealed sequences encoding proteins with a greater than 35% amino acid identity to *C. elegans* VMAT [16], tryptophan hydroxylase [17], and the serotonin receptors SER-4 [28], SER-7 [19] and MOD-1 [31] (Table 1) which are therefore strong candidates as *G.pallida* functional orthologues of these key determinants of serotonergic signalling. In this study we focussed on further characterisation of SER-7 and MOD-1 given their known importance in *C. elegans* in coupling environmental food cues to behavioural outputs relating to foraging and feeding [19,31]. In *C. elegans* a number of these genes are subject to alternate splicing and this may also occur in *G. pallida* however for the purposes of this study we focused on cloning the longest variant of each transcript. Therefore, we cannot discount the possibility that other, shorter, variants may exist.

We confirmed the functional role of each of the proteins encoded by the *G. pallida* genes identified from this bioinformatic screen using a 'model-hopping' approach with *C. elegans* i.e. we used *C. elegans* as model system to define the function of the *G. pallida* genes of interest through the generation of *C. elegans* transgenics expressing the *G. pallida* genes. For this we made use of *C. elegans* mutant strains carrying null mutations in each of the respective genes and for which there are characteristic behavioural or pharmacological phenotypes. We compared the ability of the wild-type *C. elegans* gene with that of the *G. pallida* putative orthologue to rescue the mutant phenotype. For each of the *G. pallida* genes we investigated, encoding

VMAT, tryptophan hydroxylase, SER-7 and MOD-1, we were able to reinstate wild-type behaviour in the mutant. In effect, we recapitulated *G. pallida* serotonergic signalling in *C. elegans*.

We proceeded to investigate the relevance of each of these components of the serotonergic pathway to the impact of reserpine on root invasion behaviour. We have previously shown that RNAi has non-specific toxicity to *G. pallida* stylet thrusting which confounds the use of this experimental approach to assign gene function to root invasion behaviour [25]. Therefore, we made use of the transgenic *C. elegans* strains expressing *G. pallida tph-1*, *ser-7* and *mod-1* to validate pharmacological blockers that could then be used to test for a role of each of these proteins in root invasion behaviour.

Wild-type *C. elegans* exposed to the selective tryptophan hydroxylase blocker CPA [26] showed a reduced rate of pharyngeal pumping in the presence of food, an effect which phenocopied the low pharyngeal pumping rate of the null *tph-1* mutant *mg280*. We observed a similar effect of CPA in *C. elegans* expressing *gpa-tph-1* thus indicating that CPA might be used as a pharmacological tool to interrogate the functional role of TPH in *G. pallida*. Indeed, exposure of *G. pallida* to CPA reduced fluoxetine-induced stylet thrusting and root invasion by J2s consistent with a pivotal role for serotonin signalling in these behaviours.

We found that methiothepin is a potent inhibitor of *G. pallida* stylet thrusting and root invasion. Given the very potent inhibition of stylet thrusting by methiothepin we also tested whether or not it would inhibit hatching of J2s from cysts as, like root invasion, this is also a stylet activity-dependent behaviour [5]. In contrast to the lack of effect of reserpine, methiothepin potently inhibited hatching. This discrepancy may be explained by a different ability of reserpine and methiothepin to diffuse into the cyst and across the eggshell to gain access to the unhatched J2. The effect of methiothepin on hatching is intriguing as the J2s encased within the eggs still appear viable following the methiothepin exposure. We showed that both *G. pallida* SER-7 and MOD-1 are sensitive to methiothepin, similar to the *C. elegans* receptors [29,31], and these are strong candidates for mediating the inhibitory effect of methiothepin on hatching, fluoxetine stimulated stylet thrusting and root invasion. However, methiothepin is known to act on other biogenic amine receptors, including SER-4 [45], and we cannot rule out a role for this receptor. Further studies screening for selective antagonists of *G. pallida* SER-7 and MOD-1 has the potential to provide pharmacological tools that can more precisely address this question.

An outstanding question is where serotonin acts in the neural circuits of the nematode to bring about a methiothepin-sensitive activation of stylet thrusting and root invasion. This behaviour is finely tuned to the nematode's environment and occurs at precise times during development triggered by cues from the rhizosphere or the plant which specifically engage sensory receptors and downstream signalling pathways to activate, or possibly disinhibit, stylet thrusting. By analogy with all other cue-dependent behaviours characterised in nematodes to date, the circuit which controls PPN stylet thrusting will comprise morphologically specialised sensory neurons to detect environmental cues and will connect through interneurons that integrate signals from diverse inputs, which in turn output to motorneurons to directly regulate muscles controlling the stylet. There is a good precedent that comparison between *C. elegans* and other species of nematode provides insight into neuronal circuitry [46,47]. In support of this, antibody staining for serotonin has identified putative serotonergic neurons in *Pratylenchus penetrans* in an anterior position suggesting they may be equivalent to *C. elegans* neurons ADF and NSM [12]. Further studies are required to map these pathways in *G. pallida* and overlay the expression of serotonin receptors. In *C. elegans ser-7* and *mod-1* are widely expressed consistent with the coordinating neurohormonal role of serotonin particularly with respect to feeding, foraging and energy homeostasis; it will be interesting to see whether a similar pattern plays out in *G. pallida* [30]. Taken together these observations are consistent with

the interpretation that methiothepin acts as an antagonist of either, or both, SER-7 and MOD-1 to block serotonin signalling in a circuit that is essential for engaging stylet thrusting in response to environmental stimuli to initiate hatching and root invasion.

In conclusion, we have characterised a serotonergic signalling pathway in *G. pallida* that is pivotal to root invasion behaviour and identify new molecular targets in this pathway for crop protecting chemicals.

## Materials and methods

### *Globodera pallida* maintenance and culture

Cysts of *G. pallida* (Pa2/3; population Lindley) were extracted from infested sand/loam following growth of host 'Desiree' potato plants. Dry cysts were treated with 0.1% malachite green solution for 30 mins for an initial sterilisation step, followed by extensive washing in tap water. Cysts were then incubated in an antibiotic cocktail [48] at 4 °C overnight and washed five times with sterile tap water. To induce hatching, cysts were placed in a solution of 1 part potato root diffusate to 3 parts tap water. Root diffusate was obtained by soaking washed roots of three week-old potato plants in tap water at 4 °C overnight at a rate of 80g root/litre. The diffusate was filter-sterilised before use. Significant numbers of J2 typically began hatching 1 week after rehydration in the presence of potato root diffusate at 20°C. Only J2 that had hatched within the previous 24 h were used for experiments. Prior to the experiments the J2s were washed in ddH$_2$O to remove potato root diffusate.

### *Caenorhabditis elegans* strains and culture

*C. elegans* were grown on Nematode Growth Medium (NGM) plates seeded with *Escherichia coli* (*OP50* strain) at 20°C according to standard protocols [49]. N2 (Bristol strain) *C. elegans* were employed as wild-type. GR1321 is a strain carrying a deletion in *mg280* allele for *tph-1* (ZK1290.18) and a 9.8kb deletion in *vs166* for *cam-1*. *cam-1* encodes a receptor tyrosine kinase of the immunoglobulin superfamily. This strain displays 15 phenotypes and was used in the original studies on *C. elegans tph-1* gene characterisation [17]. Another strain, MT14984, contains a deletion only in the *tph-1* gene and displays one phenotype: reduced pharyngeal pumping (CGC, made by Dan Omura). However, this strain was not available to order from CGC at the time when the experiments were carried out. GR1321 was outcrossed with N2 *C. elegans* four times (CGC database) and additionally twice more in our laboratory. *ser-7(tm1325)* strain DA2100 carries a 742 bp deletion and 38 bp insertion and displays 5 phenotypes, one of which is reduced pharyngeal pumping in response to serotonin. It was outcrossed with N2 *C. elegans* ten times (CGC database). *cat-1* encodes a vesicular monoamine transporter (VMAT) and is an orthologue of human VMATs. *cat-1(ok411)X C. elegans* mutant strain RB681 has a 429 bp deletion and is predicted to be a functional null (Wormbase). MT9668 *mod-1* (*ok103*) *V* is a null allele due to the molecular nature of the mutation, covering the entire genomic sequence corresponding to the *mod-1* gene [31]. This strain was outcrossed with N2 *C. elegans* six times (CGC database). *mod-1* encodes a serotonin-gated chloride channel.

Transgenic *C. elegans* (described below) were always assayed in parallel with positive and negative controls for the pharyngeal pumping phenotype variants i.e. on the same day with N2 and mutant strain *C. elegans*, respectively.

### Drugs and chemicals

Serotonin creatinine sulphate monohydrate (serotonin), methiothepin and 4-chloro-DL-phenylalanine methyl ester hydrochloride (CPA) were purchased from Sigma Aldrich (Dorset,

UK). Fluoxetine hydrochloride was purchased from Enzo Life Sciences (Exeter, UK) and serpasil phosphate (reserpine) from Novartis (Surrey, UK). Agar plates containing compounds were prepared as follows: For serotonin creatinine sulphate monohydrate the compound was added to 60˚C NGM and stirred to give a 10 mM final concentration. The NGM was then used to pour plates. For fluoxetine and CPA the compounds were dissolved in M9. 500μl of each compound was evenly pipetted over the surface of a 6 cm plate containing 10 ml NGM to give the final desired concentration. The plates were allowed to dry for 2 h before use. Serotonin and fluoxetine plates were stored at 4˚C for 1–2 weeks. Serpasil phosphate (reserpine) plates were prepared in a similar manner except the compound was dissolved in double distilled water and then diluted in HEPES buffer (pH 7.4). For stylet thrusting assays drugs were added to ddH$_2$O (0.1% BSA) buffered to pH 7.4 to achieve the desired concentration. Methiothepin plates were prepared as described for the rest of the drugs except 10 μM dissolved in water was added to 10 ml of NGM. These drug plates were prepared freshly for use on the day of each experiment.

### Root invasion assays

J2s of *G. pallida* were first sterilised with 0.1% chlorhexidine digluconate, 0.5 mg/ml CTAB for 25 mins and washed three times with sterile 0.01% Tween-20. The J2s were then incubated with gentle agitation for 24 h in water, 100 μM reserpine phosphate, 100 μM methiothepin or 100 μM CPA prior to inoculation of potato hairy root cultures with 25 J2s per infection point and 3 infection points per root system. Hairy root cultures were generated using *Agrobacterium rhizogenes* R1000 and multiplied as previously described [50]. Individual root systems of equivalent size growing on 9 cm plates containing Murashige and Skoog basal medium (Duchefa, Suffolk, UK) with 2% sucrose were selected for inoculation. Nine to 11 replicate plates were used for each treatment or control. Roots were stained with acid fuchsin 13 days after infection as described previously [51] and the number of nematodes in each root system counted. Each complete invasion assay was carried out on two separate occasions.

### *G. pallida* dispersal assays

Dispersal assays were performed on 5 cm plates filled with 10 ml of 2% agarose. For the reserpine assays J2s were pre-soaked in either reserpine or water for 18 h. On the day of the assay 100 μl of potato root diffusate (PRD) was spread onto the plates and the plates were sealed with Parafilm. A grid with six equal concentric circles 9, 18, 27, 36, 45 and 54 mm in diameter was placed under the assay plates and the centre was marked with a dot. Around 50–100 *G. pallida* J2s were pipetted with 5 μl of ddH$_2$O onto the centre (origin), the plate was re-sealed and the total number of J2s was counted as soon as the liquid had absorbed. The number of J2s in each concentric circle was counted after 1 and 2 h with '0' being the innermost circle and '5' the outermost.

### *G. pallida* stylet thrusting assays

Stylet thrusting assays were conducted in 20 mM HEPES pH 7.4. J2s were pipetted into 3 ml of the test solutions in 30 mm Petri dishes, mounted on the stage of an inverted microscope for viewing and the number of stylet thrusts per minute was counted at various time points as stated in figure legends. A single movement of the stylet knob forwards and then backwards to its original position was counted as one stylet thrust. Control assays were conducted in the presence of either 20 mM HEPES alone or 20 mM HEPES with drug vehicle. All assays were conducted at room temperature (20-22˚C). All drug solutions were made up on the day of use. Each solution contained 0.01% bovine serum albumin to prevent J2s from sticking to the Petri

dish. For experiments in which J2s were pre-treated with reserpine phosphate or CPA, unless stated otherwise, the J2s were pre-soaked in the drug solutions for 24 h.

## Characterisation of *C. elegans* and *G. pallida tph-1, ser-7, cat-1 and mod-1*

*G. pallida* putative orthologues of *C. elegans tph-1* (WBGene00006600), *ser-7* (WBGene00004780), *cat-1* (WBGene00000295) and *mod-1* (WBGene00003386) were first identified by BLASTP searches of the predicted protein dataset (May 2012) at http://www.sanger.ac.uk/cgi-bin/blast/submitblast/g_pallida followed by reciprocal searches of the *C. elegans* protein set at wormbase.org. [4]. Each predicted *G. pallida* gene was located in the draft genome assembly after a BLASTN search of the scaffolds via the *G. pallida* BLAST server as above. The predicted gene models were manually assessed for concordance with the mapped transcripts and primers were designed to amplify the complete predicted coding regions (corrected where necessary).

The primary amino acid sequences of *C. elegans* and *G. pallida* CAT-1, TPH-1, SER-7 and MOD-1 proteins were aligned using the UNIPROT Clustal Omega program.

Clustal-Omega uses the HHalign algorithm and its default settings as its core alignment engine [52].

## Cloning of *C. elegans* and *G. pallida tph-1, ser-7, cat-1* and *mod-1*

RNA was extracted from mixed stages *C. elegans* and from J2 *G. pallida* using an RNeasy Mini Kit (Qiagen, UK) according to the manufacturer's instructions. cDNA was reverse transcribed from 200 ng (*C. elegans*) or 500 ng (*G. pallida*) of total RNA using Superscript III reverse transcriptase with oligo-dT primers (Life Technologies, UK). *C. elegans tph-1a*, *ser-7a* and *mod-1a* were then amplified by PCR from 2 μl of cDNA using proof reading Phusion DNA polymerase (Thermo Scientific, UK), followed by addition of 1 μl of Taq DNA polymerase (Promega, UK), and continued with 10 min at 72˚C. *G. pallida* predicted orthologues were amplified from 1 μl of cDNA using Platinum HiFi polymerase (Life Technologies, UK). All primers are detailed below (Table 2), with ATG start codons underlined when present within the forward primer. PCR products were analysed by agarose gel electrophoresis and cloned into pCR8/GW/TOPO vector according to the manufacturer's instructions (Life Technologies, UK). TOP10 chemically competent cells (Life Technologies, UK) were transformed with TOPO reaction and plated onto spectinomycin (100 mg/ml) selective plates overnight at 37˚C. *C. elegans cat-1* cDNA in the pDONR201 vector (W01C8.6) was purchased from the *C. elegans* library (GE Healthcare Dharmacon, Thermo Fisher Scientific Biosciences, GmbH) and used as template to amplify and clone *cat-1a* as described for the other genes. The orientation of the cloned genes

**Table 2. Primers used for amplification of cDNA from *C. elegans* and *G. pallida*.** These primers were used to amplify cDNA for genes of interest as indicated in the left-hand column of the table.

| Gene | Forward primer | Reverse primer |
|---|---|---|
| *ce ser-7a* | AATGGCCCGTGCAGTCAACATATC | GCTAGACGTCACTTGCTTCGTGAC |
| *gpa ser-7* | GTGCCCTAATGGTCTGTCGG | CCCAAGCTTTGGGTTCAGCATGCTATTTG |
| *ce tph-1a* | TATGGATTCGTTGTTTCAGATG | CACGGAAACTCAAACTACAGG |
| *gpa tph-1* | GTAAAAATGGCTTCCGGCATG | CACTTCAATTAGTTGAAATAG |
| *ce cat-1a* | TATGTCGTACATTCTTGATTGGATC | CTAAAATGCACTGGTTGCAGAG |
| *gpa cat-1* | ATTAAAATGGCCCAATGGTTG | TTTTGGAAGCGTTTGTTGTGC |
| *ce mod-1a* | ATGAAGTTTATTCCTGAAATCACAC | TCACTGATAGTTTTGATCGAAAC |
| *gpa mod-1* | GTGAATTAATTCGCTTCCTCTC | GTGATTTGCGATGGCTGACTG |

was confirmed by digests with restriction enzymes. Subsequently, colonies that contained genes in 5' to 3' orientation were sequenced by a commercial company (Eurofins Genomics, UK). All *C. elegans* sequences subsequently used were in agreement with published data.

The *C. elegans Psnb-1* promoter was digested from the pBK3.1 vector [53] with HindIII and XbaI enzymes. pDEST vector (Life Technologies, UK) was digested with the same restriction enzymes and the fragments for *Psnb-1* promoter and linear pDEST vector were purified out of the agarose gel. The *Psnb-1* promoter was ligated into the pDEST vector in a 3:1 ratio with T4 DNA ligase (Promega, UK) overnight at 4°C and transformed into One Shot ccdB Survival 2 T1R Competent Cells (Life Technologies) grown on ampicillin (100 μg/μl) and chloramphenicol (30 μg/ml) selective plates. DNA insertion was confirmed by restriction digests and clones sequenced by Eurofins Genomics, UK. All sequenced vectors contained the predicted sequence for *Psnb-1*.

The *mod-1* promoter was amplified using the following primers: *FwPmod-1* 5'- TCGAG AAGCTTCATGTTTCACGGAACG -3'; *RvPmod-1* 5'- ACTACTCTAGAAATTTTCTTTCA CC -3', and cloned via HindIII and XbaI site into the pDEST expression vector pWormGate (https://www.sciencedirect.com/science/article/pii/S0378111905003100?via%3Dihub). MOD-1 encoding sequences were cloned into the PCR 8/GW/TOPO TA (Invitrogen) vector, and individually recombined with the PDEST vector using Gateway LR Clonase reaction. The *bona fide* sequences were verified in two independent clones by Sanger sequencing method (eurofins Genomics).

The *in vitro* recombination between an entry pCR8/GW/TOPO clone containing *C. elegans* or *G. pallida tph-1*, *ser-7*, *cat-1*, *mod-1* and a destination vector pDEST;*Psnb-1* or pDEST; *Pmod-1* as indicated, was performed using Gateway LR Clonase II enzyme mix (Life Technologies, UK) according to the manufacturer's instructions to generate expression clones for *C. elegans* microinjections. The resulting plasmids were propagated in TOP10 cells grown on ampicillin selective plates (100 μg/ml) and confirmed by restriction digests and sequencing.

### *C. elegans* transgenic experiments

*tph-1(mg280)*, *ser-7(tm1325)* and *cat-1(ok411)* one day old adult hermaphrodite *C. elegans* were injected with plasmids to drive expression of either *C. elegans* or *G. pallida tph-1*, *ser-7* or *cat-1* from a synaptobrevin promoter, *Psnb-1*, which drives expression in all neurons. This has the advantage of circumventing the need to clone the predicted native promoter for each gene of interest with the caveat that there will be ectopic expression in non-serotonergic circuits. The final expression vector contains 3'UTR terminator sequences corresponding to 3'UTR of the muscle myosin *C. elegans* gene *unc-54*, as well an optimized *C. elegans* intron to enhance expression.

*mod-1(ok103) C. elegans* were injected with a plasmid to drive expression of either *C. elegans* or *G. pallida mod-1* from the native promoter *Pmod-1*. The plasmids were injected at 30 ng μl$^{-1}$, except for *Pmod-1* which was injected at 10 ng μl$^{-1}$. Transformed *C. elegans* were identified by co-injecting L3785 (*Pmyo-3::gfp*) plasmid (50 ng μl$^{-1}$) (a gift from Andrew Fire), which drives expression of green fluorescent protein (GFP) from the body wall muscle promoter *Pmyo-3* [54]. The co-injected *gfp* transformation marker forms an extra-chromosomal array with the plasmids carrying the gene sequence and thus *C. elegans* with fluorescent green body wall muscle can be identified as carrying the plasmid of interest. For all the experiments, at least two independently transformed stable lines of transgenic *C. elegans* expressing *C. elegans* or *G. pallida tph-1*, *ser-7*, *cat-1* or *mod-1* were assayed. Results for the independent lines for each construct were in good agreement and the data presented are the pooled data from these independent lines.

## Pharmacological characterisation of the transgenic *C. elegans* strains in pharyngeal pumping assays

Experiments were performed on one day old age synchronised *C. elegans* by picking L4 stage a day before the assay. Due to the translucent nature of the nematode, pharyngeal pumping may be scored in the intact animals by counting the movements of the grinder in the terminal bulb: one complete up and down motion is counted as a single pharyngeal pump. The number of pharyngeal pumps was counted on *E. coli* OP50 in N2, in the mutants *tph-1 (mg280)*, *ser-7 (tm1325)*, *cat-1 (ok411)*, *mod-1(ok103)* and in mutants with ectopic expression of *C. elegans* (*ce*) or *G. pallida* (*gpa*) *tph-1*, *ser-7* or *cat-1* in all of the neurons (*Psnb-1* promoter).

To test the effects of CPA (4-chloro-DL-phenylalanine methyl ester hydrochloride) on pharyngeal pumping one day old adult *C. elegans* for N2, *tph-1 (mg280)* and *tph-1-1 (mg280)* with ectopic expression of *C. elegans* (*ce*) or *G. pallida* (*gpa*) *tph-1* were placed onto *E. coli* OP50 on NGM plates containing CPA and the number of pharyngeal pumps on food per min was scored after 2 h. Pharyngeal pumping of *ser-7* mutants expressing either *C. elegans* or *G. pallida ser-7* was tested after 20 min exposure to 10 mM serotonin in the absence of food. Additionally, an intact pharynx containing a terminal bulb was dissected from the rest of the nematode with a razor blade. The pharyngeal preparation was placed into 3 ml of Dent's saline (140 mM NaCl, 10 mM HEPES, 10 mM D-glucose, 6 mM KCl, 3 mM $CaCl_2$, 1 mM $MgCl_2$, pH 7.4 with 1 mM NaOH with 0.1% BSA) and the drugs serotonin or methiothepin) at a range of concentrations as indicated, in a 25 mm Petri dish. Pharyngeal pumps were scored visually, as for the intact nematodes, for 1 min. For controls, the pharyngeal pumping was scored in Dent's saline (0.1% BSA) without drugs with appropriate vehicle controls. To test the effects of fluoxetine one day old *C. elegans* for N2, *cat-1 (ok411)* and *cat-1 (ok411)* with ectopic expression of *C. elegans* (*ce*) or *G. pallida* (*gpa*) *cat-1* were placed onto unseeded NGM plates containing fluoxetine at a range of concentrations as indicated and the number of pharyngeal pumps per min was scored after 1 h.

## Pharmacological characterisation of the transgenic *C. elegans* strains in thrashing paralysis assay

In order to study the functionality of both *C. elegans* and *G. pallida* MOD-1 channels we examined the pharmacological response of MOD-1 to serotonin, using a thrashing paralysis assay as described by Ranganathan et al [31]. Briefly, 10 to 20 animals (L4+1 day stage) were placed in 200 μl of 33 mM serotonin dissolved in M9 buffer in 96-well microtitre wells. The serotonin resistance was scored, observing the swimming behaviour, every minute for a total time of 20 min. An animal was considered immobile if it did not exhibit any swimming motion for a period of 5 s.

To test whether methiothepin was able to block the serotonin-induced paralysis, we carried out the thrashing paralysis assay as described above, but with a pre-incubation with or without 10 μM methiothepin for 120 min on an NGM plate with *E. coli* OP50 bacteria before placing the *C. elegans* in the wells of the microtitre plate with serotonin. For the methiothepin treatment group, 10 μM methiothepin was included in the wells of the microtitre plate. Wild type (N2), *mod-1 (ok103)*, and the transgenic strains expressing *gpa-mod-1* in *mod-1(ok103)* were tested.

## *G. pallida* hatching assays

*G. pallida* cysts were washed in $ddH_2O$ and individual cysts were transferred to wells in a 24 well plate, containing tap water 1:3 PRD or drug solutions made using 1:3 PRD. A vehicle

control was performed for each experiment. Methiothepin was dissolved in 100% ethanol and were added to a solution of 1:3 PRD to give a final ethanol concentration of 0.5%. The cysts were soaked in 1:3 PRD solution in the presence of drug, methiothepin or reserpine, for up to 25 days. During this period hatched J2s were counted and removed from the wells. The solution in which the cysts were soaked was replaced each time a count was taken. The cysts were then removed from the drug solution and transferred to the wells of another 24 well plate containing 1:3 PRD alone to assess hatching recovery. J2 hatching was then counted in the same manner. Throughout each individual experiment the same PRD batch was used. Cumulative hatch of J2s per cyst was plotted over time. At the conclusion of the hatching experiment, cysts were transferred to ddH$_2$O and cracked open with a razor to count the number of unhatched eggs per cyst.

## Statistical analysis

Data points in graphs are presented as the mean ± standard error of the mean for the number of observations as shown in individual figures. 'N' is defined where relevant. Each experiment was repeated on at least 3 independent occasions, unless stated otherwise. Data were plotted using GraphPad Prism 7.01software (San Diego, California). Statistical significance was determined either by unpaired Student's t-test, one-way or two-way ANOVA as appropriate; significance level set at $P < 0.05$, followed by Bonferroni multiple comparisons as appropriate. A summary of the statistical analyses is provided in S1 Table. EC$_{50}$ values with 95% confidence intervals were determined using GraphPad Prism 7.01 by plotting log concentration agonist against response and fitting the data to the equation; Y = Bottom + (Top-Bottom)/(1+10^ ((LogEC50-X))).

## Supporting information

**S1 Fig. Alignment of amino acid sequences of *G. pallida* (gp) and *C. elegans* (ce) CAT-1.** UNIPROT, CLUSTAL Omega program, accessed on 08/08/2014. Identity between *C. elegans* and *G. pallida* (cloned based on gene model GPLIN_000654600) CAT-1a is 51.8%, similar positions 95. *—identical positions,: and.—similar positions.
(DOCX)

**S2 Fig. Alignment of amino acid sequences of *G. pallida* (gp) and *C. elegans* (ce) TPH-1.** UNIPROT, CLUSTAL Omega program, accessed on 14/12/2014. Cloned Gp-TPH-1 is based on *G. pallida* gene model GPLIN_000790300. Identity across all sequences is 56.4%. Identical positions 308, similar positions 79. Identity between *G. pallida* TPH-1 and *C. elegans* TPH-1a is 63.7%. *—identical positions,: and.—similar positions.
(DOCX)

**S3 Fig. Amino acid identity between the *C. elegans* (ce) and *G. pallida* (gp) serotonin receptor, SER-7.** Alignment of amino acid sequences of *G. pallida* (gp) and *C. elegans* (ce) SER-7. In the alignment shown '*' indicates identical amino acids, ':' and '.' indicate similar amino acids at each position. The identity between *G. pallida* SER-7 and *C. elegans* SER-7a is 33.2%. CLUSTAL O(1.2.1) multiple sequence alignment (10/09/2015).
(DOCX)

**S4 Fig. Alignment of amino acid sequences of *G. pallida* (Gpa) and *C. elegans* (Ce) MOD-1.** The cloned *G. pallida mod-1* is based on gene model GPLIN_001254300. In the alignment shown 'I' indicates identical amino acids, ':' and '.' indicate similar amino acids at each position. The identity between *G. pallida* MOD-1 and *C. elegans* MOD-1a is 46.1%. There are

55.7% similar positions.
(DOCX)

**S1 Table. Summary of statistical analyses.**
(XLSX)

## Author Contributions

**Conceptualization:** Peter E. Urwin, Vincent O'Connor, Lindy Holden-Dye.

**Data curation:** Lindy Holden-Dye.

**Formal analysis:** Anna Crisford, Fernando Calahorro, Jessica M. C. Marvin, Catherine J. Lilley, Vincent O'Connor, Lindy Holden-Dye.

**Funding acquisition:** Peter E. Urwin, Vincent O'Connor, Lindy Holden-Dye.

**Investigation:** Anna Crisford, Fernando Calahorro, Elizabeth Ludlow, Jessica M. C. Marvin, Jennifer K. Hibbard, Catherine J. Lilley, James Kearn, Francesca Keefe, Peter Johnson, Rachael Harmer.

**Methodology:** Anna Crisford, Fernando Calahorro, Elizabeth Ludlow, James Kearn, Francesca Keefe, Peter Johnson, Rachael Harmer.

**Project administration:** Lindy Holden-Dye.

**Resources:** Jennifer K. Hibbard, Lindy Holden-Dye.

**Supervision:** Peter E. Urwin, Vincent O'Connor, Lindy Holden-Dye.

**Validation:** Lindy Holden-Dye.

**Writing – original draft:** Anna Crisford, Fernando Calahorro, Catherine J. Lilley, Peter E. Urwin, Vincent O'Connor, Lindy Holden-Dye.

**Writing – review & editing:** Anna Crisford, Fernando Calahorro, Catherine J. Lilley, Peter E. Urwin, Vincent O'Connor, Lindy Holden-Dye.

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
