## [Decision Letter · Decision Letter 0]

1 Jun 2020

Dear Holden-Dye,

Thank you very much for submitting your manuscript "Identification and characterisation of serotonin signalling in the potato cyst nematode Globodera pallida reveals new targets for crop protection" for consideration at PLOS Pathogens. As with all papers reviewed by the journal, your manuscript was reviewed by members of the editorial board and by several independent reviewers. In light of the reviews (below this email), we would like to invite the resubmission of a significantly-revised version that takes into account the reviewers' comments.

Overall the reviewers were complementary of the work reported in this manuscript and recognized the significant contribution this research makes to the field. There were some concerns that need to be addressed. Reviewer 2 brought up a major concern regarding the connection between serotonin and infection. This concern could be addressed either by experimentation, as the reviewer mentions, or by adjusting the language. Reviewer 3 had concerns regarding the number of replicates for some of the experiments, the use of hairy root for invasion assays instead of root system assays, and some concerns regarding statistical analyses.

We cannot make any decision about publication until we have seen the revised manuscript and your response to the reviewers' comments. Your revised manuscript is also likely to be sent to reviewers for further evaluation.

Sincerely,

Adler R. Dillman, Ph.D.

Guest Editor

PLOS Pathogens

P'ng Loke

Section Editor

PLOS Pathogens

Kasturi Haldar

Editor-in-Chief

PLOS Pathogens

orcid.org/0000-0001-5065-158X

Michael Malim

Editor-in-Chief

PLOS Pathogens

orcid.org/0000-0002-7699-2064

Overall the reviewers were complementary of the work reported in this manuscript and recognized the significant contribution this research makes to the field. There were some concerns that need to be addressed. Reviewer 2 brought up a major concern regarding the connection between serotonin to infection. This concern could be addressed either by experimentation, as the reviewer mentions, or by adjusting the language. Reviewer 3 had concerns regarding the number of replicates for some of the experiments, the use of hairy root for invasion assays instead of root system assays, and some concerns regarding statistical analyses.

Reviewer's Responses to Questions

**Part I - Summary**

Reviewer #1: Manuscript PPATHOGENS-D-20-00790 describes the characterisation of components of the serotonin signalling pathway in the potato cyst nematode G. pallida. The authors adopt a novel approach combining chemical genetics and “model-hopping” to describe in considerable detail a pathway that has remained largely unexplored in any plant-parasitic nematode. Further, they demonstrate that disruption of serotonin signalling has a major impact on several aspects of the parasite life style “pre-biotrophy” (i.e. hatching and host entry). The manuscript is well written, the results unambiguous and well described (if a little repetitive), and the conclusions well justified by the data. I recommend the manuscript is accepted for publication following the most minor of revisions.

Reviewer #2: The authors present their analysis of the serotonergic nervous system of the potato cyst nematode Globodera pallida. They accomplish this through a combination of pharmacological analysis and rescue of mutations in homologous genes of C. elegans. They find that G. pallida uses a similar/identical molecular pathway to regulate the serotonergic system as found in C. elegans. While the findings do not uncover novel mechanisms specific to serotonin signaling in PPNs and the authors have already demonstrated the utility of using C. elegans as a system to investigate gene function of PPNs (Costa et al., 2009), they do make a convincing argument that investigation of this component of the PPN nervous system will lead to insight into parasitism. Overall, I find this a very nice contribution to the field that will be of broad interest.

Generally, I find the claims properly placed in the context of previous literature and well-supported by the data. The only large concern I have is in relation to their conclusions regarding nematode movement and host entry (detailed below). This could be changed by focusing more on the stylet thrusting behavior. Also, while it would enhance the story to see expression (antibody or FISH) for the proteins or genes discussed, this is not absolutely needed for the particular claims. I also have relatively minor editorial suggestions. Overall, the manuscript is very well written.

Reviewer #3: In this manuscript, Crisford and colleagues showed that reserpine inhibits serotonergic signaling that is essential for nematode stylet activation and root invasion. Authors further identified the components of serotonin signaling pathway in PCN G. pallida. Since genetic transformation of PPN is not yet established, they functionally characterized serotonin signaling components (VMAT, TPH, SER-7, MOD-1) by complementation of corresponding C. elegans mutants. Finally, authors used chemical inhibitors to confirm the role of serotonin signaling components on parasitism and root invasion.

**Part II – Major Issues: Key Experiments Required for Acceptance**

Reviewer #1: (No Response)

Reviewer #2: My main critique revolves around the conclusion tying serotonin to infection. While I generally agree with their conclusions regarding stylet thrusting, in the abstract the authors claim that impairment of host entry following exposure to reserpine is due to serotonergic signaling. From my reading this connection is based on pharmacological inhibition assays without the corresponding pharmacological rescues as were done for the stylet thrusting data (which are very convincing). For example, the authors show that incubation in reserpine reduces the number of J2 that can infect the roots and the ability of J2s to move. They also show that exposure to methiothepin also causes a reduction in nematodes able to enter the root system. However, both compounds can act outside of serotonin transport and signaling. If reserpine is producing its effect through serotonin transport alone, then application of exogenous serotonin should rescue the reserpine phenotypes shown in Figure 1, similar to the stylet thrusting behaviors in Figure 2.

Along these lines, the authors conclude that “consistent with previous reports that monoaminergic transmission, specifically that involving serotonin, is required for motility of plant-parasitic nematodes” They cite a review article by the senior author and an article by Masler et al as support. However, the Masler article found that exposure to serotonin caused a reduction in movement in both H. glycines and M. incognita (in contrast to McClure and Mende cited in the review). Horvitz et al found a similar inhibition of movement in response to exogenous serotonin application in C. elegans. This suggests that serotonin is inhibitory to movement. I would then expect reserpine to increase movement of G. pallida. If their conclusions are correct, then exposure to serotonin alone should increase the motility of J2s or at least have no effect on mobility. An alternative explanation is that reserpine is affecting multiple circuits in G. pallida (for example, dopamine). Either the authors should conduct these additional experiments or change the writing to focus on the stylet thrusting behavior.

Reviewer #3: Overall, I am supportive of this interesting study as it provides a new approach to control PPN. However, I have some concerns regarding the data presentation and interpretation.

Figure 1: I am not sure why authors used hairy root for invasion assays? The results can be misleading. Why not use root system derived from internode cuttings? I also understood that invasion assays were performed in two independent replicates. This is unacceptable. At least, three independent experiments should be performed. This is valid for the entire manuscript.

- Why to use different concentrations for figure 1A (100 uM) and 1B (50 uM)? It might be interesting to test different concentrations of reserpine for their effect on invasion.

- Serotonin signaling components seem to be well conserved across different species of nematodes. So, I suggest performing invasion assays with different species of PPNs. This should support its role to control a broad of PPN.

Figure 2A: I suggest drawing an arrow in synaptic cleft pointing towards re-uptake of serotonin by its transporter. This should facilitate readers.

Figure 3: Authors have performed ANOVA followed by Bonferroni’s multiple comparisons. However, only selective differences have been shown through asterisks. Can you please provide a lettering showing how different genotypes differ from each other? This is important to validate the claims that pharyngeal pumping movements were restored to wild type levels (N2) in complementation lines (Line 201). At the moment I just see that complementation lines were different from mutants (cat-1). This is valid for rest of the figures.

**Part III – Minor Issues: Editorial and Data Presentation Modifications**

Reviewer #1: Below I provide a series for the authors consideration:

152 – “Reserpine potently blocked the stylet response to fluoxetine but not the response to serotonin (Fig 2D) consistent with an interpretation in which the response to fluoxetine requires the presence of correctly stored vesicular serotonin which is depleted by the VMAT-blocking action of reserpine (see Fig 2A).”

A lovely set of controls

161 - “Thus, the inhibition of stylet thrusting by reserpine is sustained for at least 22 h following its removal.”

An interesting result, and a shame the full duration of the effect wasn’t delineated/predicted.

Figure 2A

Consider improving the quality, clarity, and quantity of information in this panel.

Table 1

For gene absence, I suggest the authors additionally check the genome of the related species G. rostochiensis because it is a more faithful representation of the gene content of Globodera.

Reviewer #2: It isn’t clear why a pan-neuronal promoter was used for most of the rescues except mod-1. While the data clearly shows rescue, it would be more convincing if endogenous promoters were used. I don’t think this experiment is necessary, but it would be valuable to include an explanation for the use of a pan-neuronal promoter and possible implications for data interpretation in the conclusion.

The Author Summary is primarily about reserpine and its botanical origin. This makes me think that the article will be about snakeroot and reserpine; however, reserpine is only the lead to the story and snakeroot is not featured at all. I would recommend rewriting to focus on PPNs and their behavior/neurobiology.

There are occasional changes between British and American spelling (neurone vs neuron). I’m guessing the copy editor will find these, but thought I’d mention.

Similarly, the authors move between nematodes and “worms.” I recommend sticking to nematodes or J2s.

Line 139: Seems to be missing a noun.

Line 251: Using a behavioral assay does not directly validate the ability of CPA to block TPH-1 expression. This could be done by measuring 5-HT levels. Perhaps change to “As an inhibitor of TPH, we predicted that CPA would inhibit pumping in C. elegans…”

Line 302-303: What is meant by bona fide serotonin receptor? Does this mean that serotonin has the greatest affinity for it compared to other biogenic amines? If so, then wouldn't expression in COS cells and e-phys studies be important? I recommend toning this writing down.

Line 448: I’m not sure what a ‘model-hopping’ approach is. When I Googled this it came up with electron transport in solid materials.

Lines 518: What is the goal of drying cysts and treating with 0.1% malachite green? Is there a reference for this?

The methods for the dispersal assay state there were concentric circles. I’m not sure I understand this. If it’s like I envision, then why wasn’t data shown for how far away from the origin the nematodes traveled in Figure 1B? I think this would be valuable rather than just how many left the origin.

Lines 619-621: Are the primers used for amplification the ones listed in the table below. It wasn’t clear. Are the ATGs underlined in the table the start codons?

Did the final plasmid injected include a 3’ UTR derived from C. elegans or elsewhere? Did it include an artificial intron? These are often helpful in C. elegans expression. If the pan-neuronal promoter was used because the endogenous ones didn’t work, perhaps this is the reason.

Provide a citation or source for pWormGate.

Line 678: Were these one day old adult hermaphrodites?

Figure 2A: I thought the cartoon was helpful. It may be useful to include the other compounds such as methiothepin and list the serotonin receptors on this.

Figure 8D: This doesn’t look like a DIC image, more like standard brightfield. If it is, then DIC should be spelled out.

Reviewer #3: What does n indicate in figure legends? Is this pooling of data points across the replicates? This should be clarified.

I also recommend plotting of graphs in a way showing spread of data as in Figure 8 A and 8B.

PLOS authors have the option to publish the peer review history of their article (what does this mean?). If published, this will include your full peer review and any attached files.

Reviewer #1: No

Reviewer #2: No

Reviewer #3: Yes: Shahid Siddique
---

## [Editor Report · Decision Letter 1]

14 Aug 2020

Dear Holden-Dye,

We are pleased to inform you that your manuscript 'Identification and characterisation of serotonin signalling in the potato cyst nematode Globodera pallida reveals new targets for crop protection' has been provisionally accepted for publication in PLOS Pathogens.

Best regards,

Adler R. Dillman, Ph.D.

Guest Editor

PLOS Pathogens

P'ng Loke

Section Editor

PLOS Pathogens

Kasturi Haldar

Editor-in-Chief

PLOS Pathogens

orcid.org/0000-0001-5065-158X

Michael Malim

Editor-in-Chief

PLOS Pathogens

orcid.org/0000-0002-7699-2064

Thank you for your detailed response to reviewers. The revised manuscript is improved and represents a significant advance for the field. Congratulations on fine work!
---

## [Editor Report · Acceptance letter]

28 Sep 2020

Dear Holden-Dye,

We are delighted to inform you that your manuscript, "Identification and characterisation of serotonin signalling in the potato cyst nematode Globodera pallida reveals new targets for crop protection," has been formally accepted for publication in PLOS Pathogens.

Best regards,

Kasturi Haldar

Editor-in-Chief

PLOS Pathogens

orcid.org/0000-0001-5065-158X

Michael Malim

Editor-in-Chief

PLOS Pathogens

orcid.org/0000-0002-7699-2064